# Improving Neural Optimal Transport via Displacement Interpolation

**Jaemoo Choi**
Georgia Institute of Technology
`jchoi843@gatech.edu`

**Yongxin Chen**
Georgia Institute of Technology
`ychen3148@gatech.edu`

**Jaewoong Choi**[*]
Sungkyunkwan University
`jaewoongchoi@skku.edu`

## Abstract

Optimal Transport (OT) theory investigates the cost-minimizing transport map that moves a source distribution to a target distribution. Recently, several approaches have emerged for learning the optimal transport map for a given cost function using neural networks. We refer to these approaches as the OT Map. OT Map provides a powerful tool for diverse machine learning tasks, such as generative modeling and unpaired image-to-image translation. However, existing methods that utilize max-min optimization often experience training instability and sensitivity to hyperparameters. In this paper, we propose a novel method to improve stability and achieve a better approximation of the OT Map by exploiting displacement interpolation, dubbed *Displacement Interpolation Optimal Transport Model (DIOTM)*. We derive the dual formulation of displacement interpolation at specific time $t$ and prove how these dual problems are related across time. This result allows us to utilize the entire trajectory of displacement interpolation in learning the OT Map. Our method improves the training stability and achieves superior results in estimating optimal transport maps. We demonstrate that DIOTM outperforms existing OT-based models on image-to-image translation tasks.

## 1 Introduction

Optimal Transport (OT) problem (Villani et al., 2009; Peyré et al., 2019) explores the problem of finding the cost-optimal transport map that transforms one probability distribution (*source distribution*) into another (*target distribution*). Recently, there has been a growing interest in directly learning the optimal transport map using neural networks. Throughout this paper, we call these approaches as the **OT Map**. OT Map has been widely applied across various machine learning tasks by appropriately defining the source and target distributions, such as generative modeling (Rout et al., 2022; Choi et al., 2023; 2024a; Liu et al., 2022; Lipman et al., 2023), image-to-image translation (Korotin et al., 2023; Fan et al., 2022), and domain adaptation (Flamary et al., 2016). OT Map is particularly well-suited for unsupervised (unpaired) distribution transport problems, as it enables the transport of one distribution into another using only a predefined cost function, without requiring data pairs.

Despite their potential, existing OT Map methods often encounter significant challenges in training stability. In particular, the OT models utilizing max-min objectives exhibit unstable training dynamics and sensitivity to hyperparameters (Makkuva et al., 2020; Fan et al., 2022; Rout et al., 2022; Flamary et al., 2016). These challenges limit their applicability to high-dimensional data. To address these instability issues, various approaches have been explored, such as introducing additional regularization terms in the learning objective (Rout et al., 2022; Roth et al., 2017) and generalizing the standard OT problem to the unbalanced optimal transport problem (Choi et al., 2023; 2024b).

In this paper, we propose a novel approach for learning the optimal transport map based on displacement interpolation. We refer to our model as the ***Displacement Interpolation Optimal Transport***

---

[*]Corresponding Author

***Model (DIOTM)***. We identify a fundamental connection between the (static) optimal transport map and displacement interpolation. Motivated by this relationship, we derive a max-min formulation of displacement interpolation, involving the optimal transport map and the time-dependent Kantorovich potential. Our experimental results demonstrate that DIOTM achieves more stable convergence and superior accuracy in approximating OT maps compared to existing methods. In particular, DIOTM achieves competitive FID scores in image-to-image translation tasks, such as 5.27 for Male→Female ($64 \times 64$), 7.40 for Male→Female ($128 \times 128$), and 10.72 for Wild→Cat ($64 \times 64$), comparable to the state-of-the-art results. Our contributions can be summarized as follows:

- We propose a method to learn the optimal transport map based on displacement interpolation.
- We derive the dual formulation of displacement interpolation and utilize this to formulate a max-min optimization problem for the transport map and potential function.
- We introduce a novel regularizer, called the HJB regularizer, derived from the optimality condition of the potential function.
- Our model significantly improves the training stability and accuracy of existing OT Map models that leverage min-max objectives.

**Notations and Assumptions**  Let $\mathcal{X}$ be a connected bounded convex open subspace of $\mathbb{R}^d$, and let $\mu$ and $\nu$ be absolutely continuous probability distributions with respect to Lebesgue measure. We regard $\mu$ and $\nu$ as the source and target distributions. For a measurable map $T$, $T_{\#}\mu$ represents the pushforward distribution of $\mu$. $\Pi(\mu, \nu)$ denote the set of joint probability distributions on $\mathcal{X} \times \mathcal{Y}$ whose marginals are $\mu$ and $\nu$, respectively. $c(x, y)$ refers to the transport cost function defined on $\mathcal{X} \times \mathcal{Y}$. Throughout this paper, we consider $\mathcal{X} = \mathcal{Y} \subset \mathbb{R}^d$ with the quadratic cost, $c(x, y) = \alpha\|x - y\|^2$, where $d$ indicates the dimension of data. Here, $\alpha$ is a given positive constant. Moreover, we denote $W_2(\cdot, \cdot)$ as the 2-Wasserstein distance of two distributions.

## 2 BACKGROUND

In this section, we provide a brief overview of Optimal Transport theory. These results, especially the dual formulation and displacement interpolation, will play a crucial role in our proposed method.

**Optimal Transport**  The Optimal Transport (OT) problem investigates the optimal way to transport the source distribution $\mu$ to the target distribution $\nu$ (Villani et al., 2009). The optimality of the transport plan is defined as the minimization of a given cost function. Initially, Monge (1781) formulated the OT problem with a *deterministic transport map* $T$ where $T_{\#}\mu = \nu$. However, the Monge OT problem is non-convex and the optimal transport map $T^\star$ may not exist depending on $\mu$ and $\nu$. To overcome this problem, Kantorovich (1948) introduced the following convex formulation:

$$C(\mu, \nu) := \inf_{\pi \in \Pi(\mu, \nu)} \left[ \int_{\mathcal{X} \times \mathcal{X}} c(x, y) d\pi(x, y) \right], \tag{1}$$

where the minimization is conducted over the set of joint probability distribution $\pi \in \Pi(\mu, \nu)$. We refer to this $\pi$ as the *transport plan* or *coupling* between $\mu$ and $\nu$. When the optimal transport map $T^\star$ from the Monge OT exists, the optimal coupling $\pi^\star$ satisfies $\pi^\star = (Id \times T^\star)_{\#}\mu$. For a general cost function $c(\cdot, \cdot)$ that is lower semicontinuous and lower bounded, the Kantorovich OT problem (Eq. 1) can be reformulated as follows (Villani et al. (2009), Chapter 5):

$$C(\mu, \nu) = \sup_{V \in L^1(\nu)} \left[ \int_{\mathcal{X}} V^c(x) d\mu(x) + \int_{\mathcal{X}} V(y) d\nu(y) \right], \tag{2}$$

where the potential function $V \in L^1(\nu)$ is an integrable function with respect to $\nu$ and the $c$-transform of $V$ is defined as

$$V^c(x) = \inf_{y \in \mathcal{Y}} \left( c(x, y) - V(y) \right). \tag{3}$$

This formulation (Eq. 2) is called the *semi-dual formulation of OT*.

Rout et al. (2022) and Fan et al. (2022) proposed a method for learning the optimal transport map $T^\star$ by leveraging this semi-dual formulation (Eq. 2) of OT and applied this $T^\star$ for generative modeling. In generative modeling, the source distribution $\mu$ and the target distribution $\nu$ correspond to the

Gaussian prior and the target data distribution, respectively. Specifically, these models parametrize the potential $V = V_\phi$ in Eq. 2 and the transport map $T_\theta : \mathcal{X} \to \mathcal{Y}$ as follows:

$$T_\theta : x \mapsto \arg\min_{y \in \mathcal{X}} [c(x, y) - V_\phi(y)] \quad \Leftrightarrow \quad V_\phi^c(x) = c(x, T_\theta(x)) - V_\phi(T_\theta(x)). \quad (4)$$

Note that the parametrization of $T_\theta$ on the left-hand side is equivalent to the representation of $V_\phi^c$ on the right-hand side, by the definition of the $c$-transform (Eq. 3). Based on this, we arrive at the following optimization problem:

$$\mathcal{L}_{V_\phi, T_\theta} = \sup_{V_\phi} \left[ \int_{\mathcal{X}} \inf_{T_\theta} [c(x, T_\theta(x)) - V_\phi(T_\theta(x))] \, d\mu(x) + \int_{\mathcal{X}} V_\phi(y) d\nu(y) \right]. \quad (5)$$

Intuitively, $T_\theta$ and $V_\phi$ serve similar roles as the generator and the discriminator of a GAN (Goodfellow et al., 2020). A key difference is that $T_\theta$ is trained to minimize the cost $c(x, T_\theta(x))$, since $T_\theta$ learns the optimal transport map. For convenience, we denote the optimization problem $\mathcal{L}_{V_\phi, T_\theta}$ (Eq. 5) as the OT-based generative model (OTM) (Fan et al., 2022).

**Dynamic Optimal Transport and Displacement Interpolation**  In this paragraph, we provide a **close connection between the *dynamic optimal transport problem* and *displacement interpolation***. While the (static) optimal transport (Eq. (1)) focuses solely on how each $x \sim \mu$ is transported to $y \sim \nu$, the dynamic optimal transport problem tracks the continuous evolution of $\mu$ to $\nu$. Formally, the dynamic formulation of the Kantorovich OT problem (Eq. 1) for the quadratic cost $c(x, y) = \alpha \|x - y\|^2$ can be expressed as follows:

$$\inf_{v:[0,1] \times \mathcal{X} \to \mathcal{X}} \left[ \int_0^1 \int_{\mathcal{X}} \alpha \|v_t(x)\|^2 \rho_t(x) dx dt; \ \frac{\partial \rho_t}{\partial t} + \nabla \cdot (v_t \rho_t) = 0, \ \rho_0 = \mu, \ \rho_1 = \nu \right]. \quad (6)$$

This dynamic formulation of OT (Eq. 6) is called the *Benamou-Brenier formulation (Benamou & Brenier, 2000)*. Note that the dynamic transport plan $\{\rho_t\}_{0 \le t \le 1}$ evolves from $\mu$ to $\nu$, and this evolution is governed by the ODE $dx/dt = v_t(x)$ through the continuity equation.

Interestingly, the optimal solution of this dynamic OT problem has a simple form. Along each ODE trajectory $\{x(t) \mid dx/dt = v_t(x), x(0) = x_0\}$, the velocity field $v$ remains constant. Moreover, when the deterministic optimal transport map $T^\star$ exists, the following holds:

$$\rho_t^{dis} := [(1-t) \cdot Id + t \cdot T^\star]_\# \mu \qquad \text{and} \qquad \rho_t^{dis} = \rho_t^\star \qquad \text{for } 0 \le t \le 1. \quad (7)$$

where $Id$ denotes the identity map and $\rho^\star$ denotes the optimal dynamic OT plan of Eq. 6. $\rho_t^{dis}$ is called *McCann's Displacement Interpolation (DI)* (McCann, 1997). Eq. 7 shows that the pushforward of linear interpolation between $Id$ and the static OT map $T^\star$ is equivalent to the dynamic OT plan $\rho_t^\star$. Hence, throughout this paper, we simply denote the displacement interpolation as $\rho_t^\star$. Furthermore, it is well known that the displacement interpolants satisfy the following property (Theorem 7.21 in Villani et al. (2009)):

$$\rho_t^\star = \arg\inf_\rho \mathcal{L}_{DI}(t, \rho) \quad \text{where} \quad \mathcal{L}_{DI}(t, \rho) = (1-t) W_2^2(\mu, \rho) + t W_2^2(\rho, \nu). \quad (8)$$

Note that $\mathcal{L}_{DI}$ corresponds to the Wasserstein-2 barycenter problem between the two probability distributions $\mu, \nu$ (Agueh & Carlier, 2011; Kolesov et al., 2024b). In other words, Eq. 8 represents the equivalence between the displacement interpolants and the Wasserstein barycenter. This equivalence will be utilized in Sec 3 to derive our approach to neural optimal transport, i.e., learning the optimal transport map $T^\star$ with a neural network. We establish how the optimal potential and transport maps for each $\rho_t^\star$ are related and use this relationship to improve neural optimal transport.

## 3   METHOD

In this section, we propose our method, called the *Displacement Interpolation Optimal Transport Model (**DIOTM**)*. Our model leverages displacement interpolation to improve the stable estimation of the optimal transport map using neural networks. In Sec 3.1, we derive our two main theorems for deriving our learning objective. In Sec 3.2, we describe how we implement our DIOTM model based on these theoretical results.

## 3.1 DUAL FORMULATION OF DI AND THE RELATIONSHIP BETWEEN INTERPOLATION POTENTIAL FUNCTIONS

In this subsection, we provide two theoretical results: the **Dual Formulation of Displacement Interpolation** (Thm 3.1) and the **Relationship between Interpolation Potential Functions** (Thm 3.3). Thm 3.1 will be utilized to derive our main learning objective (Eq. 17) and Thm 3.3 will be employed to formulate our regularizer (Eq. 19).

**Dual Problem of Displacement Interpolant** Here, we begin with the dual formulation of the minimization characterization of the interpolant $\rho_t^\star$ (Eq. 8). The dual problem can be expressed as Eq. 9 in the following Theorem 3.1. Note that while the primal formulation (Eq. 8) optimizes over the set of probability distribution $\rho$, the dual formulation optimizes over two potential functions $f_{1,t}, f_{2,t}$.

**Theorem 3.1.** *Given the assumptions in Appendix A, for a given $t \in (0,1)$, the minimization problem $\inf_\rho \mathcal{L}_{DI}(t, \rho)$ (Eq. 8) is equivalent to the following dual problem:*

$$\sup_{f_{1,t}, f_{2,t} \in \mathcal{C} \text{ with } (1-t)f_{1,t}+tf_{2,t}=0} \left[ (1-t)\int_\mathcal{X} f_{1,t}^c(x)d\mu(x) + t\int_\mathcal{Y} f_{2,t}^c(y)d\nu(y) \right]. \quad (9)$$

*where the supremum is taken over two continuous potential functions $f_{1,t} : \mathcal{Y} \to \mathbb{R}$ and $f_{2,t} : \mathcal{X} \to \mathbb{R}$, which satisfy $(1-t)f_{1,t}+tf_{2,t} = 0$. Note that the assumptions in Appendix A guarantee the existence and uniqueness of displacement interpolants $\rho_t^\star$, the forward optimal transport map $\overrightarrow{T}_t^\star$ from $\mu$ to $\rho_t^\star$, and the backward transport map $\overleftarrow{T}_t^\star$ from $\nu$ to $\rho_t^\star$. Based on this, we have the following:*

$$\overrightarrow{T}_t^\star(x) \in \mathrm{arginf}_{y \in \mathcal{Y}} \left[ c(x,y) - f_{1,t}^\star(y) \right], \quad \overleftarrow{T}_t^\star(y) \in \mathrm{arginf}_{x \in \mathcal{X}} \left[ c(x,y) - f_{2,t}^\star(x) \right], \quad (10)$$

*$\mu$-a.s. for $x$ and $\nu$-a.s. for $y$. By applying the same c-transfrom parametrization as in Eq. 4, we derive the following max-min formulation of the dual problem:*

$$\sup_{f_{1,t}, f_{2,t} \in \mathcal{C} \text{ with } (1-t)f_{1,t}+tf_{2,t}=0} (1-t)\int_\mathcal{X} \inf_{\overrightarrow{T}_t} \left( c(x, \overrightarrow{T}_t(x)) - f_{1,t}(\overrightarrow{T}_t(x)) \right) d\mu(x)$$

$$+ t\int_\mathcal{Y} \inf_{\overleftarrow{T}_t} \left( c(\overleftarrow{T}_t(y), y) - f_{2,t}(\overleftarrow{T}_t(y)) \right) d\nu(y). \quad (11)$$

Note that the max-min formulation above requires optimization over four functions: the two potentials $f_{1,t}, f_{2,t}$ and the two transport maps $\overrightarrow{T}_t, \overleftarrow{T}_t$. Here, **we provide an intuitive interpretation of this max-min optimization (Eq. 11)**. Two transport maps $\overrightarrow{T}_t, \overleftarrow{T}_t$ act as generators of the interpolant $\rho_t^\star$ from $\mu$ and $\nu$, respectively. Two potential functions $f_1, f_2$ serve similar roles to discriminators for the generated samples (fake samples) from $\overrightarrow{T}_t, \overleftarrow{T}_t$. The discriminator values for the true samples $\rho_t^\star$ cancels out because of the potential condition $(1-t)f_{1,t} + tf_{2,t} = 0$. Formally,

$$(1-t)\int f_{1,t}(z)\,d\rho_t^\star(z) + t\int f_{2,t}(z)\,d\rho_t^\star(z) = \int \left[(1-t)f_{1,t} + tf_{2,t}\right](z)\,d\rho_t^\star(z) = 0. \quad (12)$$

In fact, using this potential condition, **we can combine these two potentials into a single value function (or potential function) $V_t$.** This simplified formulation of Eq. 9 will be used to derive our regularizer in Theorem 3.3.

**Corollary 3.2.** *For a given $t \in (0,1)$, let $f_{1,t}(y) = tV_t(y)$ and $f_{2,t}(x) = -(1-t)V_t(x)$ for some value function $V_t : \mathcal{X} = \mathcal{Y} \to \mathbb{R}$. Then, the dual formulation of displacement interpolation (Eq. 9) can be rewritten as follows:*

$$\sup_{V_t} \left[ \int_\mathcal{X} V_t^{c_{0,t}}(x)d\mu(x) + \int_\mathcal{Y} (-V_t)^{c_{t,1}}(y)d\nu(y) \right], \quad (13)$$

*where $c_{s,t}(x,y) = \frac{\alpha\|x-y\|^2}{t-s}$ for every $0 \le s < t \le 1$.*

**Relationship between Interpolation Potential Functions** Here, based on Corollary 3.2, we derive the optimality condition, that each interpolant value function $V_t$ satisfies, as the time-dependent value function $V(t, x) = V_t(x)$. From now on, we will **denote the value function in its time-dependent form:** $V(t, x) : (0, 1) \times \mathcal{X} = \mathcal{Y} \to \mathbb{R}$.

---

**Algorithm 1** Training algorithm of DIOTM

---

**Require:** The source distribution $\mu$ and the target distribution $\nu$. Transport networks $\overrightarrow{T}_{\theta_1}$, $\overleftarrow{T}_{\theta_2}$ and the discriminator network $V_\phi$. Total iteration number $K$, and regularization hyperparameter $\lambda$.

1: **for** $k = 0, 1, 2, \ldots, K$ **do**
2:     Sample a batch $x \sim \mu$, $y \sim \nu$, $t \sim U[0, 1]$.
3:     $x_t \leftarrow (1-t)x + t\overrightarrow{T}_{\theta_1}(x)$, $y_t \leftarrow (1-t)\overleftarrow{T}_{\theta_2}(y) + ty$.
4:     Update $V_\phi$ to increase the $\mathcal{L}_\phi$

$$\mathcal{L}_\phi = -V_\phi(t, x_t) + V_\phi(t, y_t) - \lambda \mathcal{R}(V_\phi(t, x_t)) - \lambda \mathcal{R}(V_\phi(t, y_t)).$$

5:     Update $\overrightarrow{T}_{\theta_1}$ to decrease the loss: $c_{0,t}(x, x_t) - V_\phi(t, x_t)$.
6:     Update $\overleftarrow{T}_{\theta_2}$ to decrease the loss: $c_{t,1}(y_t, y) + V_\phi(t, y_t)$.
7: **end for**

---

**Theorem 3.3.** *Given the assumptions in Appendix A, the optimal $V_t^\star$ in Eq. 13 satisfies the following:*

$$V_t^\star = \arg\sup_{V_t} \left[ \int_\mathcal{X} V_t^{c_{0,t}}(x)d\mu(x) + \int_\mathcal{Y} V_t(x)d\rho_t^\star(x) \right], \tag{14}$$

*up to constant $\rho^\star$-a.s.. Moreover, there exists $\{V_t^\star\}_{0 \leq t \leq 1}$ that satisfies Hamilton-Jacobi-Bellman (HJB) equation, i.e.*

$$\partial_t V_t^\star + \frac{1}{4\alpha} \|\nabla V_t^\star\|^2 = 0, \qquad \rho^\star\text{-a.s.} \tag{15}$$

In Sec 3.2, we use this HJB optimality condition (Eq. 15) as a regularizer for the value function $V$ in our model. In Sec 5, we demonstrate that, when combined with our displacement interpolation, this regularizer significantly improves the stability of optimal transport map estimation.

## 3.2 DISPLACEMENT INTERPOLATION OPTIMAL TRANSPORT MODEL

In this subsection, we introduce our model, called *Displacement Interpolation Optimal Transport Model (DIOTM)*. The goal of our model is to learn the optimal transport map between the source distribution $\mu$ and the target distribution $\nu$. DIOTM is trained by utilizing the displacement interpolation $\rho_t^\star$ (Eq. 7) between these two distributions. In DIOTM, the forward and backward transport maps $\overrightarrow{T}$, $\overleftarrow{T}$ are trained to match all intermediate distributions. As a result, each transport map is not directly trained to generate the boundary distributions $\mu$ and $\nu$, but instead exploits the matching of intermediate distributions. This approach enables our model to achieve a more stable estimation of the optimal transport map.

**Parametrization of the DI Dual Problem** Our DIOTM learns the static optimal transport map between $\mu$ and $\nu$ by exploiting the displacement interpolation (DI) $\rho_t^\star$, which is the solution of the dynamic optimal transport problem (Villani et al., 2009). However, there are some challenges when using DI to learn the static transport map. While our goal is to learn the optimal transport maps between $\mu$ and $\nu$, the max-min dual formulation of the DI dual problem (Eq. 11) applies to each specific time $t \in (0, 1)$. In other words, the intermediate transport maps $\overrightarrow{T}_t$, $\overleftarrow{T}_t$ and the potential $V_t$ are defined separately for each $t$. Therefore, **we represent these interpolant generators $\overrightarrow{T}_t$, $\overleftarrow{T}_t$ through the boundary generators $\overrightarrow{T}$, $\overleftarrow{T}$ by incorporating the optimality condition of DI (Eq. 7).** Specifically, we parametrize the interpolant generators $\overrightarrow{T}_t$ and $\overleftarrow{T}_t$ as follows:

$$\overrightarrow{T}_t(x) = (1-t)x + t\overrightarrow{T}_{\theta_1}(x), \quad \overleftarrow{T}_t(y) = (1-t)y + t\overleftarrow{T}_{\theta_2}(y) \quad \text{for } t \in (0, 1). \tag{16}$$

where $\overrightarrow{T}_{\theta_1}$ and $\overleftarrow{T}_{\theta_2}$ parametrize the optimal transport maps from the source $\mu$ to the target $\nu$ and from $\nu$ to $\mu$, respectively. The optimality condition of DI clarifies how the intermediate optimal transport maps $\overrightarrow{T}_t^\star$, $\overleftarrow{T}_t^\star$ are related to each other. We utilize this condition to parametrize the entire interpolant $\rho_t^\star$ for $t \in (0, 1)$ using just two networks $\overrightarrow{T}_{\theta_1} : \mathcal{X} \to \mathcal{Y}$ and $\overleftarrow{T}_{\theta_2} : \mathcal{Y} \to \mathcal{X}$. Note that this optimality condition is satisfied under the optimal value function $V_\phi$ is given. Thus, this parametrization is

introduced for better efficiency. Additionally, we already investigated how the potentials, specifically the value function $V(t, x)$, are related in Theorem 3.3. Therefore, by combining Eq 13 and 16, we arrive at our main max-min learning objective:

$$\mathcal{L}_{\phi,\theta}^{DI} = \sup_{V_\phi} \int_{\mathcal{X}} \inf_{\overrightarrow{T}_{\theta_1}} \mathbb{E}_t \left[ \frac{\alpha}{t} \|x - \overrightarrow{T}_t(x)\|^2 - V_\phi(t, \overrightarrow{T}_t(x)) \right] d\mu(x)$$

$$+ \int_{\mathcal{Y}} \inf_{\overleftarrow{T}_{\theta_2}} \mathbb{E}_t \left[ \frac{\alpha}{1-t} \|\overleftarrow{T}_t(y) - y\|^2 + V_\phi(t, \overleftarrow{T}_t(y)) \right] d\nu(y). \quad (17)$$

Here, $\alpha$ indicates the cost intensity hyperparameter, i.e., $c(x, y) = \alpha\|x - y\|_2^2$. Note that the expectation with respect to $t$ is for aggregating the interpolant $\rho_t^\star$ for $t \in (0, 1)$. When represented with our transport map parametrizations $\overrightarrow{T}_{\theta_1}$ and $\overleftarrow{T}_{\theta_2}$ (Eq. 16), our main learning objective $\mathcal{L}_{\phi,\theta}^{DI}$ can be expressed as follows:

$$\mathcal{L}_{\phi,\theta}^{DI} = \sup_{V_\phi} \int_{\mathcal{X}} \inf_{\overrightarrow{T}_{\theta_1}} \mathbb{E}_t \left[ \alpha t \|x - \overrightarrow{T}_\theta(x)\|^2 - V_\phi(t, \overrightarrow{T}_t(x)) \right] d\mu(x)$$

$$+ \int_{\mathcal{Y}} \inf_{\overleftarrow{T}_{\theta_2}} \mathbb{E}_t \left[ \alpha(1-t) \|\overleftarrow{T}_\theta(y) - y\|^2 + V_\phi(t, \overleftarrow{T}_t(y)) \right] d\nu(y). \quad (18)$$

Moreover, we introduce the *HJB regularizer* $\mathcal{R}(V_\phi)$, which is derived from the HJB optimality condition of the value function, proved in Theorem 3.3:

$$\mathcal{R}(V_\phi) = \mathbb{E}_{t, x \sim \rho_t} \left| 2\alpha \, \partial_t V_\phi(t, x) + \frac{1}{2} \|\nabla V_\phi(t, x)\|^2 \right|. \quad (19)$$

As a result, the learning objective can be summarized as follows:

$$\mathcal{L}_{\phi,\theta} = \mathcal{L}_{\phi,\theta}^{DI} + \lambda \mathcal{R}(V_\phi(t, x)). \quad (20)$$

where $\lambda > 0$ denotes the HJB regularizer intensity hyperparameter.

**Algorithm** We present our training algorithm for DIOTM (Algorithm 1). Our adversarial training objective $\mathcal{L}_{\phi,\theta}$ updates alternatively between the value function $V_\phi$ and the two transport maps $\overrightarrow{T}_\theta, \overleftarrow{T}_\theta$, similar to GAN framework (Goodfellow et al., 2020). Note that we simplified Algorithm 1 by omitting the non-dependent terms for each neural network. Additionally, we apply the HJB regularizer to both generated distributions, i.e., the forward generated distribution from $\overrightarrow{T}_\theta$ and the backward generated distribution from $\overleftarrow{T}_\theta$. Finally, throughout all experiments, we use the uniform distribution for time sampling, i.e., $t \sim U[0, 1]$. However, the time sampling distribution can be freely modified. We leave the investigation of the optimal time sampling distribution to future work. In particular, if we set $t \sim \delta_1$, i.e., $t = 1$, the training algorithm becomes similar to OTM (Rout et al., 2022). In this case, we do not use any displacement interpolation information for $t \in (0, 1)$. The only difference is our HJB regularizer.

## 4 RELATED WORK

OT problem addresses a transport map between two distributions that minimizes the predefined cost function. Starting from the dual formulations (Kantorovich, 1948; Vacher & Vialard, 2022a;b; Gallouët et al., 2021), diverse methods for estimating OT Map have been developed based on minimax problem (Liu et al., 2019; Makkuva et al., 2020; An et al., 2020; Fan et al., 2022; Rout et al., 2022; Choi et al., 2023; 2024b; Korotin et al., 2023). In particular, Fan et al. (2022); Rout et al. (2022); Korotin et al. (2023) derived adversarial algorithm from the semi-dual formulation of OT problem and properly recovered OT maps compared to other previous works (Makkuva et al., 2020). Moreover, they provided moderate performance in image generation and image translation tasks for large-scale datasets. Another line of research focuses on the dynamical formulation of OT problems (Chen et al., 2021; Shi et al., 2024b; Liu et al., 2024; 2022; Neklyudov et al., 2023; Gushchin et al., 2024). Several works (Zhang & Chen, 2021; Chen et al., 2021; Shi et al., 2024b; Liu et al., 2024) use sampling-based

Table 1: **Evaluation of optimal transport map** on the synthetic datasets between OTM (Rout et al., 2022) and ours, based on the 2-Wasserstein distance $W_2$ and the L2 distance between transport maps.

| Metric | G→8G | | G→25G | | Moon→Spiral | | G→Circles | |
|---|---|---|---|---|---|---|---|---|
| | OTM | DIOTM | OTM | DIOTM | OTM | DIOTM | OTM | DIOTM |
| $W_2$ (↓) | 4.93 | **3.72** | 10.09 | **6.49** | **0.40** | 0.55 | 3.96 | **2.34** |
| L2 (↓) | 6.48 | **4.38** | 13.09 | **10.00** | 1.77 | **1.67** | 6.23 | **5.44** |

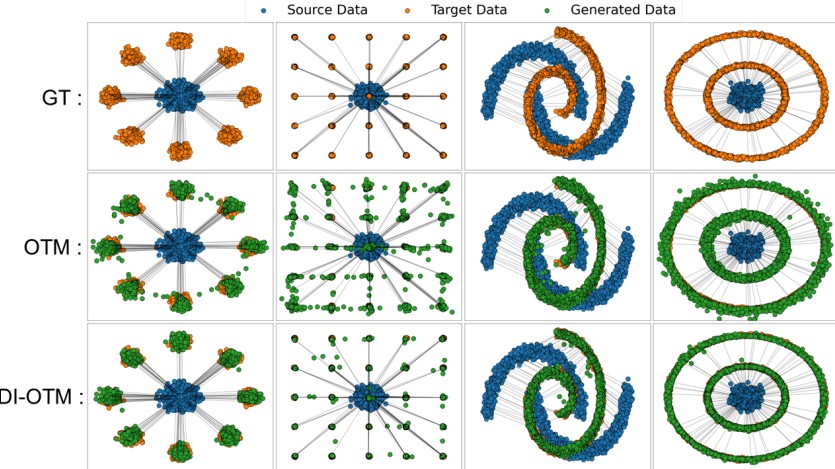

Figure 1: **Visualization of transport maps $T$ on synthetic datasets.** The transport map is visualized as a black line connecting each source sample $x$ to its corresponding generated data $T(x)$.

approaches, which require numerically simulating ODEs or SDEs during training. More recently, some methods (Neklyudov et al., 2023; Gushchin et al., 2024) have introduced simulation-free techniques, often incorporating adversarial learning strategies. Since our approach also leverages the dynamical properties of OT, we compare our method with Shi et al. (2024b); Gushchin et al. (2024), both of which have demonstrated scalability in image translation tasks.

## 5 EXPERIMENTS

In this section, we conduct experiments on various datasets to evaluate our model from the following perspectives.

- In Sec 5.1, we compare our model with the ground-truth solution from POT Flamary et al. (2021) on synthetic datasets to assess how well our model approximates OT maps.
- In Sec 5.2, we compare our model with various OT models on image-to-image translation tasks to evaluate the scalability of our model.
- In Sec 5.3, we evaluate the training stability of our DIOTM model and investigate the effectiveness of HJB regularizer compared to other approaches, such as OTM regularizer and R1 regularizer.

For implementation details of experiments, please refer to Appendix B.

### 5.1 OPTIMAL TRANSPORT MAP EVALUATION ON SYNTHETIC DATASETS

First of all, **we evaluate whether our model can accurately learn the optimal transport map $T^\star$ from the source distribution $\mu$ and the target distribution $\nu$.** We assess our model against OTM (Rout et al., 2022; Fan et al., 2022) by comparing them to the discrete OT solution from the POT library (Flamary et al., 2021). As discussed in Sec 3.2, when the time sampling distribution is set to $\delta_{t=1}$, our DIOTM presents a similar framework with OTM. Hence, we consider OTM an appropriate baseline for demonstrating the advantage of using displacement interpolation. Note that OTM demonstrated the most competitive performance as a neural optimal transport map in Korotin et al. (2022b). Also, the discrete OT solution indicates the optimal transport map between empirical

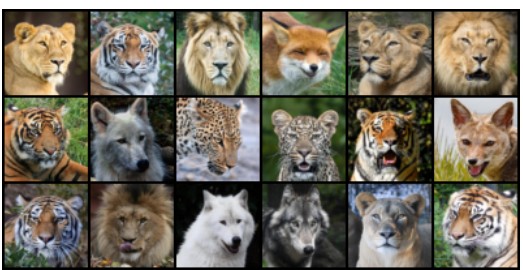 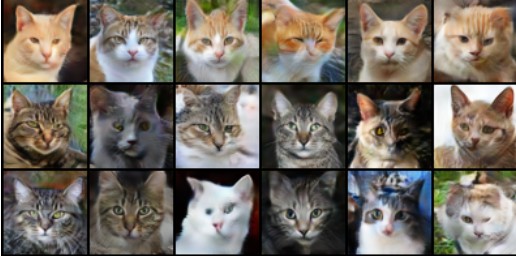

Figure 2: **Image-to-Image translation results** of DIOTM for Wild $\rightarrow$ Cat ($64 \times 64$) on AFHQ (Choi et al., 2020). The left figure shows the source images and the right figure shows the corresponding translated images by DIOTM.

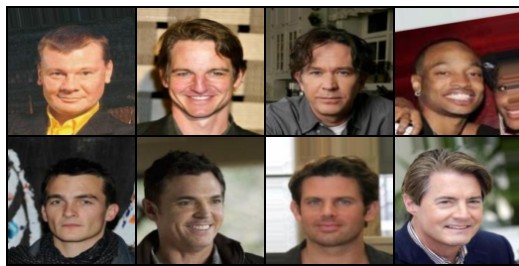 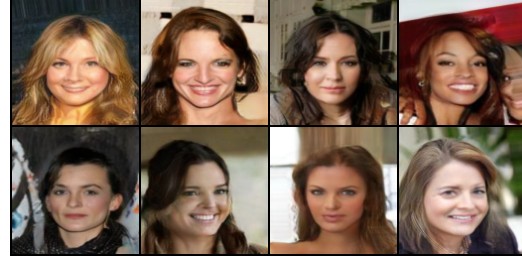

Figure 3: **Image-to-Image translation results** of DIOTM for Male $\rightarrow$ Female ($128 \times 128$) on CelebA (Liu et al., 2015).

distributions, i.e., $\mu = \sum_i \delta_{x_i}, \nu = \sum_j \delta_{y_j}$, computed via convex optimization. Hence, this discrete OT solution serves as the proxy ground-truth solution of the continuous OT map. We tested our model on four synthetic datasets: *Gaussian-to-8Gaussian (G→8G), Gaussian-to-25Gaussian (G→25G), Moon-to-Spiral, and Gaussian-to-Circles (G→Circles)*.

Fig. 1 visualizes the transport maps and Tab. 1 presents the quantitative evaluation metric results. In Tab. 1, we evaluate each transport map through two metrics. First, we calculate the 2-Wasserstein distance $W_2(\overrightarrow{T}_{\theta\#}\mu, \nu)$ between the generated distribution $\overrightarrow{T}_{\theta\#}\mu$ and the target distribution $\nu$. Second, we evaluate whether the neural optimal transport correctly recovers the optimal pairings $(x, T(x))$. Specifically, we compute the discrete optimal transport $T^\star_{disc}$ on test datasets using POT and measure the L2 distance between transport maps, i.e., $\int_{\mathcal{X}} \|\overrightarrow{T}(x) - T^\star(x)\|_2^2 d\mu_{test}(x)$. Fig. 1 shows that our DIOTM more accurately approximates the target distribution, as indicated by the smaller distribution error between the orange target data and the green generated data. This is further supported by the quantitative results. In Tab. 1, DIOTM achieves a smaller distribution error ($W_2$) on three out of four datasets and consistently better recovers the optimal coupling ($L2$) across all datasets. In summary, our DIOTM provides a better approximation of the optimal transport map $T^\star$ compared to OTM.

## 5.2 SCALABILITY EVALUATION IN IMAGE-TO-IMAGE TRANSLATION TASKS

We assessed our model on several Image-to-Image (I2I) translation benchmarks: *Male $\rightarrow$ Female (Liu et al., 2015)* ($64 \times 64$), *Wild $\rightarrow$ Cat (Choi et al., 2020)* ($64 \times 64$), and *Male $\rightarrow$ Female (Liu et al., 2015)* ($128 \times 128$). Intuitively, the optimal transport map serves as a generator for the target distribution, which maps each input $x$ to its cost-minimizing counterpart $y$. For instance, if we consider Male $\rightarrow$ Female task with the quadratic cost $\|x - y\|_2^2$, the optimal transport map translates each male image into a female image while minimizing the pixel-level difference. Consequently, various (entropic) optimal transport approaches are widely used for the I2I translation tasks. Therefore, we compared our model with the optimal transport models (NOT (Fan et al., 2022) and OTM (Rout et al., 2022)) and the entropic optimal transport models (DSBM (Shi et al., 2024a) and ASBM (Gushchin et al., 2024)) on image-to-image (I2I) translation tasks.

Fig. 2 and 3 present the I2I translation results. In each figure, the left subfigure shows source image samples. The right subfigure displays the translated images generated by our transport map. DIOTM successfully generates the target distributions (Cat and Female images) while preserving the identity

Table 2: **Image-to-Image translation benchmark** results compared to existing neural (entropic) optimal transport models. † indicates the results conducted by ourselves. DSBM scores are taken from (Gushchin et al., 2024; De Bortoli et al., 2024).

| Data | Model | FID ($\downarrow$) |
|---|---|---|
| Male→Female (64x64) | CycleGAN (Zhu et al., 2017) | 12.94 |
| | NOT (Korotin et al., 2023) | 11.96 |
| | OTM† (Fan et al., 2022) | 6.42 |
| | DIOTM† (Ours) | **5.27** |
| Wild→Cat (64x64) | DSBM (Shi et al., 2024a) | 20+ |
| | OTM† (Fan et al., 2022) | 12.42 |
| | DIOTM† (Ours) | **10.72** |
| Male→Female (128x128) | DSBM (Shi et al., 2024a) | 37.8 |
| | ASBM (Gushchin et al., 2024) | 16.08 |
| | OTM† (Fan et al., 2022) | 7.55 |
| | DIOTM† (Ours) | **7.40** |

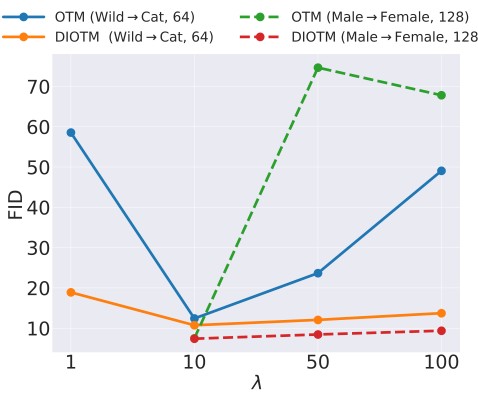

Figure 4: **Ablation study on the regularizer hyperparameter $\lambda$.**

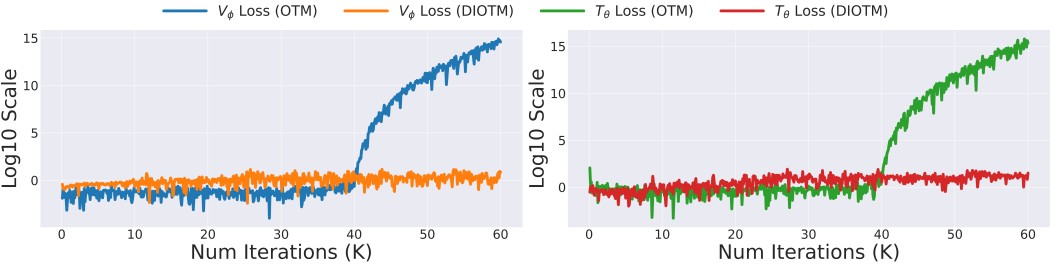

Figure 5: **Visualization of the stable training dynamics of DIOTM** on Wild→Cat ($64 \times 64$). The loss values are visualized on a $\log_{10}$ scale.

Table 3: **Comparison of our HJB regularizer with the OTM and R1 regularizers** on the DIOTM model. Our HJB regularizer exhibits superior performance and stability to $\lambda$.

| | Model | G→8G | | | | G→25G | | | | Moon→Spiral | | | |
|---|---|---|---|---|---|---|---|---|---|---|---|---|---|
| | $\lambda$ | 0.1 | 0.2 | 1.0 | 10 | 0.1 | 0.2 | 1.0 | 10 | 0.1 | 0.2 | 1.0 | 10 |
| $W_2$ ($\downarrow$) | OTM | 22.08 | 22.90 | DIV | 31.35 | 68.01 | 89.62 | DIV | 81.02 | 19.99 | 14.19 | 15.66 | 33.80 |
| | R1 | 3.59 | 5.01 | 3.29 | 4.42 | 9.20 | **9.94** | 11.78 | DIV | 1.91 | 2.08 | 1.05 | 2.74 |
| | HJB | **1.93** | **2.69** | **2.92** | **3.21** | **7.19** | 14.64 | **7.99** | **12.38** | **0.54** | **0.59** | **0.30** | **1.31** |
| L2 ($\downarrow$) | OTM | 27.41 | 28.21 | DIV | 34.47 | 96.89 | 97.98 | DIV | 87.05 | 20.96 | 15.01 | 34.31 | 33.80 |
| | R1 | 4.49 | 5.39 | 3.87 | 5.14 | 86.05 | 17.64 | 19.52 | DIV | 2.88 | 3.56 | 2.36 | 3.74 |
| | HJB | **3.05** | **3.44** | **3.36** | **3.98** | **16.51** | **15.82** | **11.11** | **15.64** | **1.42** | **2.25** | **1.13** | **2.27** |

of the input source images. In practice, our DIOTM model trains two transport maps in both directions $\overrightarrow{T}_\theta$ and $\overleftarrow{T}_\theta$. The results for the reverse image-to-image translation are included in Appendix C. Tab. 2 provides the quantitative evaluation results. We adopted the FID score (Heusel et al., 2017) for quantitative comparison. The FID score assesses whether each model accurately generates the target distribution. As shown in Tab. 2, the DIOTM model demonstrates state-of-the-art results among existing (entropic) OT-based methods on I2I translation benchmarks. Specifically, our model significantly outperforms other entropic OT-based models with a FID score of 7.40 in the higher resolution case of Male → Female ($128 \times 128$). While OTM achieved comparable results at a specific hyperparameter of $\lambda = 10$, OTM diverges for all other $\lambda \in \{50, 100\}$ with FID > 60 (Fig. 4). In contrast, our DIOTM consistently maintains a stable performance of FID < 10 for $\lambda \in \{50, 100\}$ (Fig. 4). In summary, DIOTM exhibits superior scalability in handling higher-resolution image datasets compared to previous OT Map models such as NOT and OTM.

### 5.3 FURTHER ANALYSIS

In this subsection, we provide an in-depth analysis of our DIOTM model. Specifically, we demonstrate the stable training dynamics of DIOTM, compare the HJB regularizer $\mathcal{R}(V_\phi)$ with various

regularizers introduced in other neural optimal transport models, and conduct an ablation study on the regularization hyperparameter $\lambda$.

**Stable Training Dynamics** The previous approaches to neural optimal transport with adversarial learning often suffer from training instability (Choi et al., 2024b). These models tend to diverge after long training and are sensitive to hyperparameters. In contrast, our DIOTM offers stable training dynamics. Fig. 5 visualizes the loss values for the transport map $T_\theta$ and the value function $V_\phi$ throughout the training process. Note that we visualized these loss values on a $\log_{10}$ scale. In OTM (Rout et al., 2022), the $T_\theta$ loss gradually explodes as training progresses. This unstable training dynamics has been a major challenge for the OT Map models based on minimax objectives. On the contrary, DIOTM exhibits stable loss dynamics without the abrupt divergence phenomenon, observed in OTM.

**Comparison to Various Regularization Methods** We investigate the effect of our HJB regularizer $\mathcal{R}(V_\phi)$ (Eq. 19) compared to other regularization methods. Specifically, we compare two alternatives: OTM regularizer $\mathcal{R}_{\text{OTM}}$ (Fan et al., 2022; Rout et al., 2022) and R1 regularizer $\mathcal{R}_{\text{R1}}$ (Roth et al., 2017). These regularizers are also introduced to stabilize the training value function $V_\phi$. We incorporate these regularizers into our algorithm by modifying the regularization term $\mathcal{R}$ in line 4 of Alg. 1 as follows:

- $\mathcal{R}_{\text{OTM}}(t, x_t, y_t) = \|\nabla_y (c_{0,t}(x, x_t) - V_\phi(t, x_t))\| + \|\nabla_y (c_{t,1}(y, y_t) + V_\phi(t, y_t))\|$.
- $\mathcal{R}_{\text{R1}}(t, x_t, y_t) = \|\nabla_y V_\phi(t, x_t)\|^2 + \|\nabla_y V_\phi(t, y_t)\|^2$.

We assessed these regularizers on synthetic datasets to measure the accuracy of each neural optimal transport map, as in Tab. 1. Tab. 3 provides the results of each regularization method. Note that because we are comparing different regularizers, a direct comparison under the same $\lambda$ is not meaningful. Instead, we need to focus on the best results of each regularizer and their robustness to the regularization intensity parameter $\lambda$. In Tab. 3, our HJB regularizer attains the best and stable results on all three synthetic datasets. We interpret this result by focusing that the HJB regularizer is the only regularizer that incorporates the time derivative $\partial_t V_\phi$. Our time-dependent value function $V_\phi(t, x)$ is trained to distinguish the displacement interpolation for each $t$. Therefore, regularizing the behavior of our value function across time $t$ is beneficial for DIOTM.

**Effect of the Regularization Hyperparameter $\lambda$** Finally, we conducted an ablation study on the regularization hyperparameter $\lambda$ (Eq. 20) in the image-to-image translation tasks of Wild $\rightarrow$ Cat ($64 \times 64$) and Male $\rightarrow$ Female ($128 \times 128$). Note that, unlike Tab. 3, we compare our model with OTM (Rout et al., 2022), not the DIOTM model with the OTM regularizer. Fig. 4 demonstrates that our model exhibits significantly greater stability regarding the regularization hyperparameter $\lambda$, in comparison to OTM. Specifically, in Wild $\rightarrow$ Cat ($64 \times 64$), our model maintains decent performance for $\lambda \in \{1, 50, 100\}$, showing the FID score around 20 even in the worst case. In contrast, the FID scores of OTM fluctuate severely from below 20 to around 60 depending on $\lambda$.

## 6 CONCLUSION

In this paper, we introduced the Displacement Interpolation Optimal Transport Model (DIOTM), a neural optimal transport method based on displacement interpolation. Our method is motivated by the equivalence between displacement interpolation and dynamic optimal transport. We derived the dual formulation of displacement interpolant and developed a method that utilizes the entire trajectory of displacement interpolation to improve neural optimal transport learning. Our experiments demonstrated that DIOTM achieves more accurate and stable optimal transport map estimation compared to previous method. A major limitation of this work is the requirement to train both bidirectional transport maps $\overrightarrow{T}$ and $\overleftarrow{T}$. In image-to-image translation tasks, both transport maps are meaningful, e.g. Male $\leftrightarrow$ Female. However, in generative modeling, the reverse transport map $\overleftarrow{T}$ from the data to the prior noise distribution is not always necessary. In these cases, training $\overleftarrow{T}$ can be an unnecessary cost. Another limitation of this work is that our approach is limited to the quadratic cost. This is because our displacement interpolation parametrization in Eq. 16 is only valid under the quadratic cost assumption.

ACKNOWLEDGEMENTS

Jaemoo was supported by the National Research Foundation of Korea(NRF) grant funded by the Korea government(KSIT) [RS-2024-00410661] and [RS-2024-00342044]. YC is supported by the National Science Foundation under Award No. DMS-2206576. Jaewoong was partially supported by the National Research Foundation of Korea(NRF) grant funded by the Korea government(MSIT) [RS-2024-00349646]. We thank the Center for Advanced Computation in KIAS for providing computing resources.

**Ethics Statement** In this paper, we aim to advance the field of machine learning by developing a relatively stable and high-performing algorithm. We hope that further investigation will enable our methodology to be applied as a stabilized algorithm across various machine learning applications. By providing a robust framework, we anticipate that our approach will contribute to improving the reliability and effectiveness of machine learning solutions in diverse contexts.

**Reproducibility Statement** To ensure the reproducibility of our work, we submitted the anonymized source in the supplementary material, provided complete proofs of our theoretical results in Appendix A, and included the implementation and experiment details in Appendix B.

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

## A    PROOFS

**Assumptions**    Let $\mathcal{X} \in \mathbb{R}^d$ be a closure of a connected bounded convex open subspace of $\mathbb{R}^d$. Let $\dim(\partial \mathcal{X}) < d$. Let $\mathcal{P}_{2,ac}$ be a collection of probability distributions on $\mathcal{X}$ that are absolutely continuous and have finite second moments. Let $\mu, \nu \in \mathcal{P}_{2,ac}$. Moreover, let $c(x,y) = \tau \|x - y\|^2$. Here, $\tau$ is a given positive constant.

### A.1    PROOF OF THEOREM 3.1

**Theorem A.1.** *Given the assumptions in Appendix A, for a given $t \in (0,1)$, the minimization problem $\inf_\rho \mathcal{L}_{DI}(t, \rho)$ (Eq. 8) is equivalent to the following dual problem:*

$$\sup_{f_{1,t}, f_{2,t} \text{ with } (1-t)f_{1,t}+tf_{2,t}=0} \left[ (1-t) \int_\mathcal{X} f_{1,t}^c(x) d\mu(x) + t \int_\mathcal{X} f_{2,t}^c(y) d\nu(y) \right]. \quad (21)$$

*where the supremum is taken over two potential functions $f_{1,t} : \mathcal{Y} \to \mathbb{R}$ and $f_{2,t} : \mathcal{X} \to \mathbb{R}$, which satisfy $(1-t)f_{1,t} + tf_{2,t} = 0$. Note that the assumptions in Appendix A guarantee the existence and uniqueness of displacement interpolants $\rho_t^\star$, the forward optimal transport map $\overrightarrow{T}_t^\star$ from $\mu$ to $\rho_t^\star$, and the backward transport map $\overleftarrow{T}_t^\star$ from $\nu$ to $\rho_t^\star$. Based on this, we have the following:*

$$\overrightarrow{T}_t^\star(x) \in \operatorname{arginf}_{y \in \mathcal{Y}} \left[ c(x,y) - f_{1,t}^\star(y) \right] \quad \mu\text{-a.s..} \quad (22)$$

$$\overleftarrow{T}_t^\star(x) \in \operatorname{arginf}_{x \in \mathcal{X}} \left[ c(x,y) - f_{2,t}^\star(x) \right] \quad \nu\text{-a.s..} \quad (23)$$

*Proof.* We prove the theorem by two steps. First, we prove Eq. 21 following the duality theorem in Korotin et al. (2022a) and Kolesov et al. (2024a). Then, we discuss the existence and uniqueness of the potential $f_1, f_2$ and the optimal transport maps. Using these uniquesness, we prove Eq. 22.

**Step 1.**    We first prove Eq. 21. Suppose $0 < t < 1$ is given. By applying the dual form of OT problem Kantorovich (1948), for every $\rho \in \mathcal{P}_2$, the following equation satisfies:

$$(1-t)W_2^2(\mu, \rho) + tW_2^2(\rho, \nu)$$
$$= \sup_{f_1, f_2 \in \mathcal{C}(\mathcal{X})} \left[ (1-t)\mathbb{E}_{x \sim \mu}[f_{1,t}^c(x)] + t\mathbb{E}_{y \sim \nu}[f_{2,t}^c(y)] + \mathbb{E}_{z \in \rho}[(1-t)f_{1,t}(z) + tf_{2,t}(z)] \right]. \quad (24)$$

Then, by applying Theorem 3.4 in Sion (1957), we obtain the following:

$$\mathcal{L}^* = \inf_\rho (1-t)W_2^2(\mu, \rho) + tW_2^2(\rho, \nu),$$
$$= \inf_\rho \sup_{f_{1,t}, f_{2,t} \in \mathcal{C}(\mathcal{X})} \left[ (1-t)\mathbb{E}_{x \sim \mu}[f_{1,t}^c(x)] + t\mathbb{E}_{y \sim \nu}[f_{2,t}^c(y)] + \mathbb{E}_{z \in \rho}[(1-t)f_{1,t}(z) + tf_{2,t}(z)] \right],$$
$$= \sup_{f_{1,t}, f_{2,t} \in \mathcal{C}(\mathcal{X})} \inf_\rho \left[ (1-t)\mathbb{E}_{x \sim \mu}[f_{1,t}^c(x)] + t\mathbb{E}_{y \sim \nu}[f_{2,t}^c(y)] + \mathbb{E}_{z \sim \rho}[(1-t)f_{1,t}(z) + tf_{2,t}(z)] \right]. \quad (25)$$

Note that we can swap minimax to max-min problem due to the compactness assumption of the space $\mathcal{X}$. Now, suppose $m = m(f_{1,t}, f_{2,t}) := \inf_{z \in \mathcal{X}}(1-t)f_{1,t}(z) + tf_{2,t}(z)$. Let $\tilde{f}_{1,t} = (m(f_{1,t}, f_{2,t}) - tf_{2,t})/(1-t)$. Then, $m = (1-t)\tilde{f}_{1,t}(z) + tf_{2,t}(z) \leq (1-t)f_{1,t}(z) + tf_{2,t}(z)$, which implies $\tilde{f}_{1,t} \leq f_{1,t}$. Then, we can easily obtain $f_{1,t}^c \leq \tilde{f}_{1,t}^c$. With this inequality, we can obtain the following equality:

$$\sup_{f_{1,t}, f_{2,t} \in \mathcal{C}(\mathcal{X})} \inf_\rho \left[ (1-t)\mathbb{E}_{x \sim \mu}[f_{1,t}^c(x)] + t\mathbb{E}_{y \sim \nu}[f_{2,t}^c(y)] + \mathbb{E}_{z \sim \rho}[(1-t)f_{1,t}(z) + tf_{2,t}(z)] \right],$$
$$= \sup_{f_{1,t}, f_{2,t} \in \mathcal{C}(\mathcal{X})} \left[ (1-t)\mathbb{E}_{x \sim \mu}[f_{1,t}^c(x)] + t\mathbb{E}_{y \sim \nu}[f_{2,t}^c(y)] + m(f_{1,t}, f_{2,t}) \right],$$
$$= \sup_{f_{2,t} \in \mathcal{C}(\mathcal{X})} \left[ (1-t)\mathbb{E}_{x \sim \mu}[\tilde{f}_{1,t}^c(x)] + t\mathbb{E}_{y \sim \nu}[f_{2,t}^c(y)] + m(\tilde{f}_{1,t}, f_{2,t}) \right],$$
$$= \sup_{f_{2,t} \in \mathcal{C}(\mathcal{X})} \left[ (1-t)\mathbb{E}_{x \sim \mu}\left[ \left( -\frac{tf_{2,t}}{1-t} \right)^c (x) \right] + t\mathbb{E}_{y \sim \nu}[f_{2,t}^c(y)] \right]. \quad (26)$$

Note that the last equation is obtained by using the following property: $(f_{1,t} + a)^c = f_{1,t}^c - a$ for any constant $a$. By letting $f_{1,t} = -\frac{tf_{2,t}}{1-t}$, we finally obtain Eq. 21.

**Step 2.** In this step, we prove the uniqueness of the optimal potential pair $(f_{1,t}^\star, f_{2,t}^\star)$ of Eq. 21. Let

$$\rho^\star = \arg\inf_\rho (1-t)W_2^2(\mu, \rho) + tW_2^2(\rho, \nu). \tag{27}$$

First, we should show the well-definedness of above equation. In other words, we should show the existence and uniqueness of $\rho^\star$. Due to the absolutely continuity assumption of $\mu$, there exists unique deterministic optimal transport map between $\mu$ and $\nu$ (See Chapter 1 or Theorem 9.2 of Villani et al. (2009)). Thus, by applying Corollary 7.23 in Villani et al. (2009), there exist unique solution $\rho^\star$.

Now, consider the following optimization problem:

$$\sup_{\hat{f}_{1,t}, \hat{f}_{2,t}} \left[ (1-t)\left( \int_\mathcal{X} \hat{f}_{1,t}^c(x)d\mu(x) + \int_\mathcal{X} \hat{f}_{1,t}(x)d\rho^\star(x) \right) \right.$$
$$\left. + t\left( \int_\mathcal{X} \hat{f}_{2,t}^c(y)d\nu(y) + \int_\mathcal{X} \hat{f}_{2,t}(y)d\rho^\star(y) \right) \right]. \tag{28}$$

Trivially, the first and the second term of Eq. 28 break down into two independent optimization problems. By using the Kantorovich duality theorem (See Kantorovich (1948) or Theorem 5.10 Villani et al. (2009)), the solution value of Eq. 28 is obviously $\mathcal{L}^\star$. Moreover, since the solution pair $(f_{1,t}^\star, f_{2,t}^\star)$ of Eq. 21 satisfies $(1-t)f_{1,t}^\star + tf_{2,t}^\star = 0$, it is included in the optimal potential pair of Eq. 28.

Note that we assumed that $\mu$ is absolutely continuous and the space $\mathcal{X}$ is convex, and a closure of a connected open set. Then, on the support of $\mu$, the Kantorovich potentials $\hat{f}_{1,t}$ and $\hat{f}_{2,t}$ is unique up to constant on the connected support of $\mu$ and $\nu$, respectively (See Staudt et al. (2022)). Therefore, the optimal potentials $f_{1,t}^\star, f_{2,t}^\star$ of Eq. 21 are a Kantorovich potentials of $W_2^2(\mu, \rho^\star)$ and $W_2^2(\nu, \rho^\star)$, respectively.

Finally, since $\mu \in \mathcal{P}_{2,ac}$, there exists a measurable deterministic optimal transport map $T^\star$ which transport $\mu$ to $\rho^\star$ (See Chapter 1 or Corollary 9.4 in Villani et al. (2009)). Then, by Theorem 5.10 and Remark 5.13 in Villani et al. (2009), we can easily obtain Eq. 22. $\square$

Let $f_{1,t}(x) = tV(t, x)$ and $f_{1,t}(x) = -(1-t)V(t, x)$ for every $t \in (0, 1)$. Then, we can rewrite Eq. 9 as follows:

$$\sup_{V_t} \left[ \int_\mathcal{X} V_t^{c_{0,t}}(x)d\mu(x) + \int_\mathcal{X} (-V_t)^{c_{t,1}}(y)d\nu(y) \right], \tag{29}$$

where $c_{s,t}(x, y) = \frac{\tau\|x-y\|^2}{t-s}$ for every $0 \le s < t \le 1$.

The following theorem shows the connection between the optimal potential of transport problem between $\mu$ to $\rho_t^\star$, and our potential function $V$.

**Theorem A.2.** *The optimal $V_t^\star$ in Eq. 13 satisfies the following:*

$$V_t^\star = \arg\sup_{V_t} \left[ \int_\mathcal{X} V_t^{c_{0,t}}(x)d\mu(x) + \int_\mathcal{X} V_t(x)d\rho_t^\star(x) \right], \tag{30}$$

*up to constant $\rho^\star$-a.s.. Moreover, there exists $\{V_t^\star\}_{0 \le t \le 1}$ that satisfies Hamilton-Jacobi-Bellman (HJB) equation, i.e.*

$$\partial_t V_t + \frac{1}{4\tau}\|\nabla V_t\|^2 = 0, \quad \rho^\star\text{-a.s.} \tag{31}$$

*Proof.* By the Step 2 of the proof of the Theorem A.1, we have shown that the optimal $(f_1^\star, f_2^\star)$ which solves Eq. 21 is a Kantorovich dual function of $W_2^2(\mu, \rho^\star)$ and $W_2^2(\nu, \rho^\star)$, respectively. By directly applying this fact, we obtain Eq. 30.

Now, we prove Eq. 31, the HJB equation. Let $\rho^\star$ be defined as the Step 2 of the proof of Theorem A.1. Then, as discussed in Step 2, there exists unique $\rho^\star$ that satisfies Eq. 27. Since $\mu$ is absolutely

continuous, it is well known that $\rho^\star$ is also absolutely continuous. Due to the compactness of the space $\mathcal{X}$, the optimal $V_t$ is differentiable with respect to $x \in \mathcal{X}$ $\rho^\star$-a.s.. Now, by applying Theorem 7.36 and Remark 7.37 of Villani et al. (2009), there exists $V : (0, 1) \times \mathcal{X} \to \mathbb{R}$ such that

$$-V_s(x) = \inf_y \left( c_{s,t}(x, y) - V_t(y) \right), \tag{32}$$

for all $0 < s < t < 1$. By applying Hopf-Lax formula, we obtain,

$$-V(s, X_s) = \inf_{v:[0,1]\times\mathcal{X}\to\mathcal{X}} \left[ \int_s^t \tau\|v_t\|^2 du - V_t(X_t) \right], \tag{33}$$

where $\dot{X}_t = v_t(X_t)$. Thus, by organizing the Eq. 33, we obtain Eq. 31 as follows:

$$0 = \lim_{t \to s} \inf_{v:[0,1]\times\mathcal{X}\to\mathcal{X}} \left[ \frac{1}{t-s} \int_s^t \tau\|v\|^2 - \frac{1}{t-s} \left( V_t(X_t) - V_s(X_s) \right) \right], \tag{34}$$

$$= \inf_v \left( \tau\|v_s\|^2 - \partial_s V_s - \nabla V_s \cdot v_s \right) = -\left( \partial_s V_s + \frac{\|\nabla V_s\|^2}{4\tau} \right). \tag{35}$$

Note that $\text{law}(X_t) = \rho_t^\star$. Thus, $\partial_t V_t + \frac{\|\nabla V_t\|^2}{4\tau} = 0$ $\rho_t^\star$-a.s.. □

## B    IMPLEMENTATION DETAILS

For all experiments, we parametrize $\overrightarrow{T}_\theta, \overleftarrow{T}_\theta : \mathcal{X} \times \mathbb{R}^n \to \mathcal{X}$ where $\mathcal{X}$ is a data space, and $z \in \mathbb{R}^n$ is an auxiliary variable. As reported in Choi et al. (2023; 2024b), incorporating the auxiliary variable slightly improves the performance. For all OTM experiments we have implemented, we use the same network and the same parameter to DIOTM, unless otherwise stated.

### B.1    2D EXPERIMENTS

**Data Description**    In this paragraph, we describe our synthetic datasets:

- **8-Gaussian**: For $m_i = 12 \left( \cos \frac{i}{4}\pi, \sin \frac{i}{4}\pi \right)$ where $i = 0, 1, \dots, 7$ and $\sigma = 0.4$, the distribution is defined as the mixture of $\mathcal{N}(m_i, \sigma^2)$ with an equal probability.
- **25-Gaussian**: For $m_{ij} = (8i, 8j)$ where $i = -2, -1, \dots, 2$ and $j = -2, -1, \dots, 2$, the distribution is defined as the mixture of $\mathcal{N}(m_{ij}, \sigma^2)$ where $\sigma = 0.01$.
- **Moon to Spiral**: We follow Choi et al. (2024b).
- **Two Circles**: We first uniformly sample from the circles of radius 8 and 16 with the center at origin. Then, we add Gaussian noise with standard deviation of 0.2.

**Network Architectures**    We first describe the value function network $V_\phi(t, x)$. The input $x \in \mathcal{X}$ is embedded using a two-layer MLP with a hidden dimension of 128. The time variable $t$ is embedded using a positional embedding of dimension 128, followed by a two-layer MLP, also with a hidden dimension of 128. These two embeddings are then added and passed through a three-layer MLP. The SiLU activation function is applied to all MLP layers. Besides, for the transport map networks, we employ the same network to Choi et al. (2023) with hidden dimension of 128.

**Training Hyperparameters**    We set Adam optimizer with $(\beta_1, \beta_2) = (0, 0.9)$, learning rate of $10^{-4}$ and the number of iteration of 120K. We set $\alpha = 0.1$ and $\lambda = 1$.

**Discrete OT Solver**    We used the POT library Flamary et al. (2021) to obtain an accurate transport plan $\pi_{pot}$. We used 1000 training samples for each dataset in estimating $\pi_{pot}$ to sufficiently reduce the gap between the true continuous measure and the empirical measure.

### B.2    IMAGE TRANSLATION

**Training Hyperparameters**    We follow the large neural network architecture introduced in Xiao et al. (2021). We use Adam optimizer with $(\beta_1, \beta_2) = (0, 0.9)$, learning rate of $10^{-4}$, and trained

for 60K iterations. We use a cosine scheduler to gradually decrease the learning rate from $10^{-4}$ to $5 \times 10^{-5}$. The batch size of 64 and 32 is employed for $64 \times 64$ and $128 \times 128$ image datasets, respectively. We use $\alpha = 0.001$ for CelebA image dataset, and $\alpha = 0.0005$ for AFHQ dataset. We use ema rate of 0.9999 for $64 \times 64$ image datasets and 0.999 for 128x128 image datasets.

**Evaluation Metric**  Regarding the FID computation, we followed the evaluation scheme of (De Bortoli et al., 2024) for the Wild→Cat experiments and (Korotin et al., 2023; Gushchin et al., 2024) for the CelebA experiments for a fair comparison. Specifically, in the Wild→Cat experiments, we generated ten samples for each source test image. Since the source test dataset consists of approximately 500 samples, we generated 5000 generated samples. Then, we computed the FID score with the training target dataset, which also contains 5000 samples. Also, in the CelebA experiment, we computed the FID score using the test target dataset, which includes 12247 samples. We generated the same 12247 samples and compared them with the test target dataset.

## C  ADDITIONAL QUALITATIVE RESULTS

In this section, we include some qualitative results on image-to-image translation tasks. We visualize the source images and its transported samples.

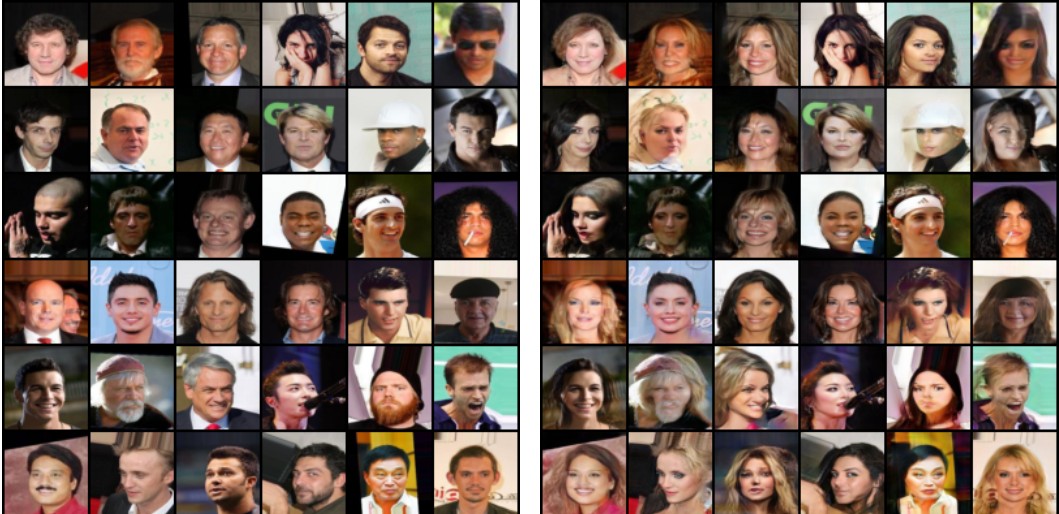

Figure 6: Unpaired Male → Female translation for $64 \times 64$ CelebA image.

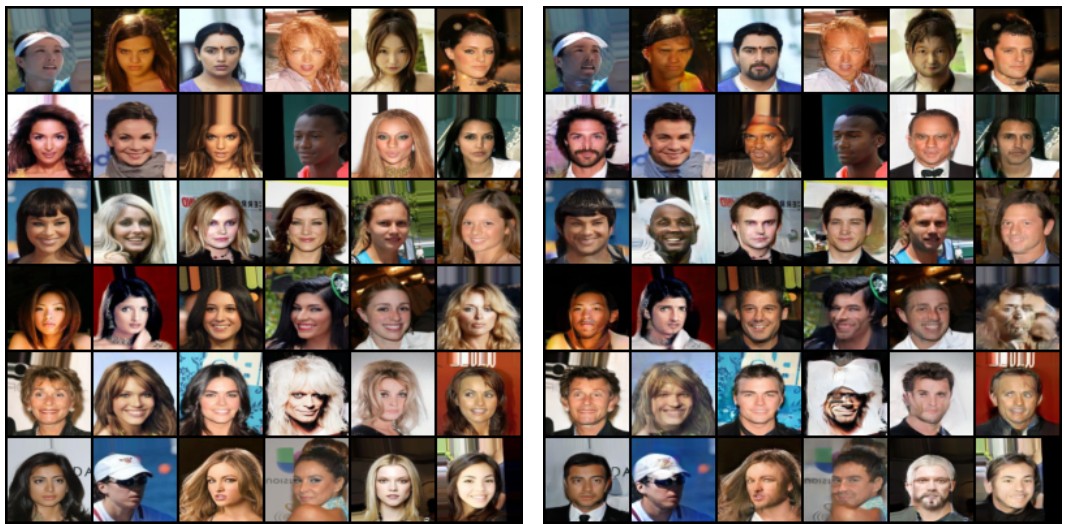

Figure 7: Unpaired Female → Male translation for 64 × 64 CelebA image.

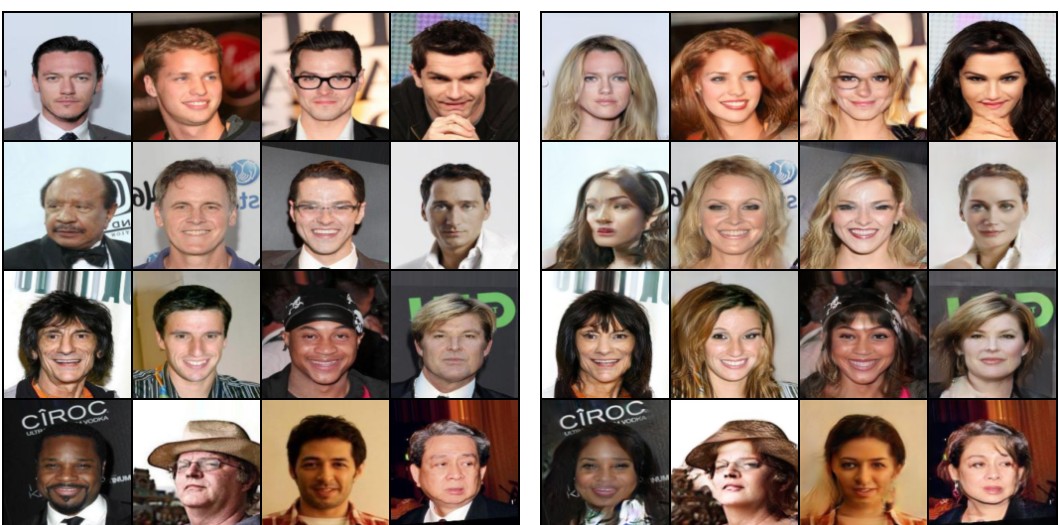

Figure 8: Unpaired Female → Male translation for 128 × 128 CelebA image.

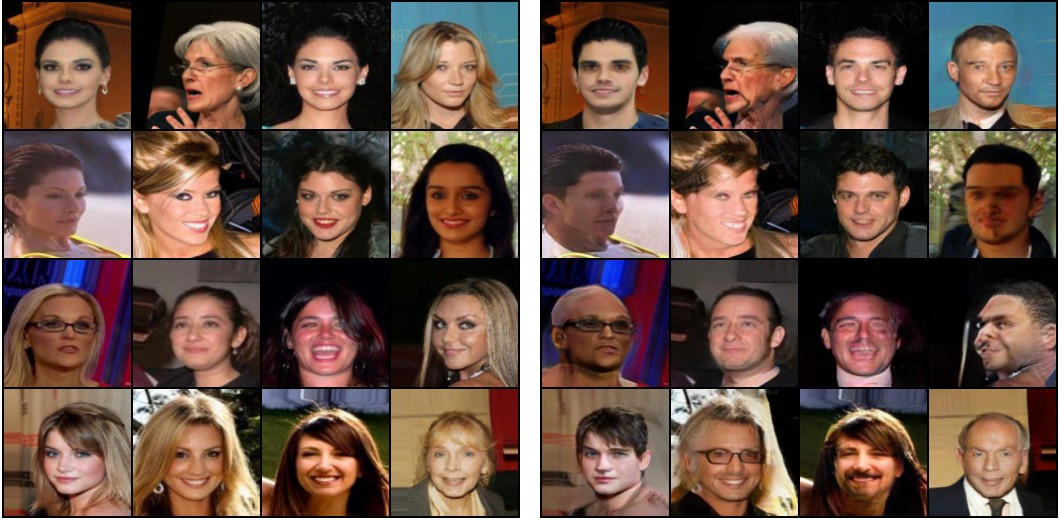

Figure 9: Unpaired Male → Female translation for 128 × 128 AFHQ image.

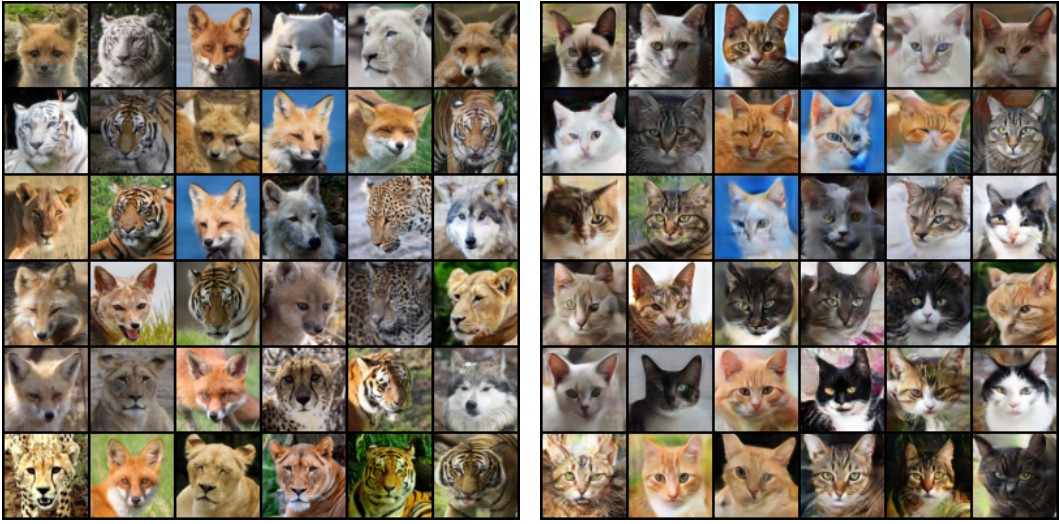

Figure 10: Unpaired Wild → Cat translation for 64 × 64 AFHQ image.

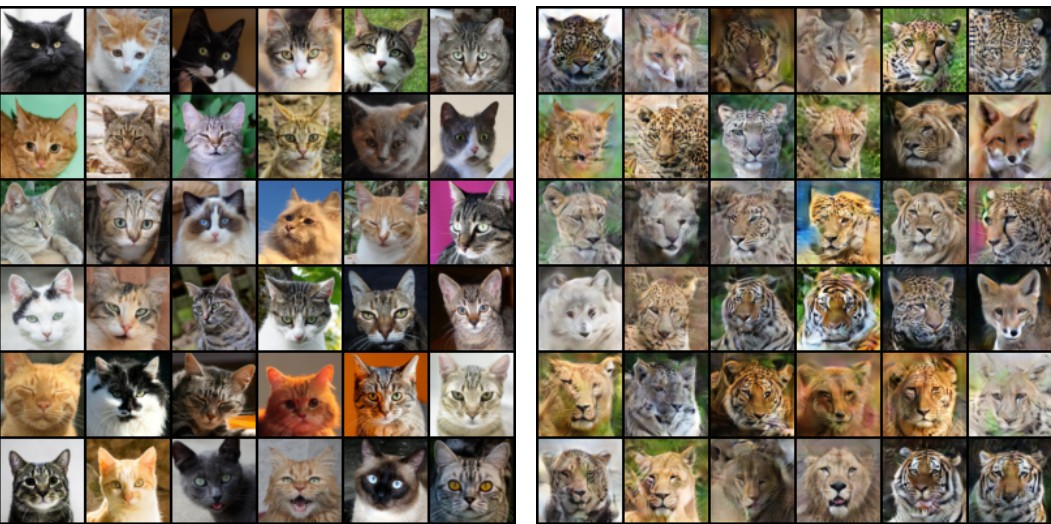

Figure 11: Unpaired Cat → Wild translation for 64 × 64 AFHQ image.

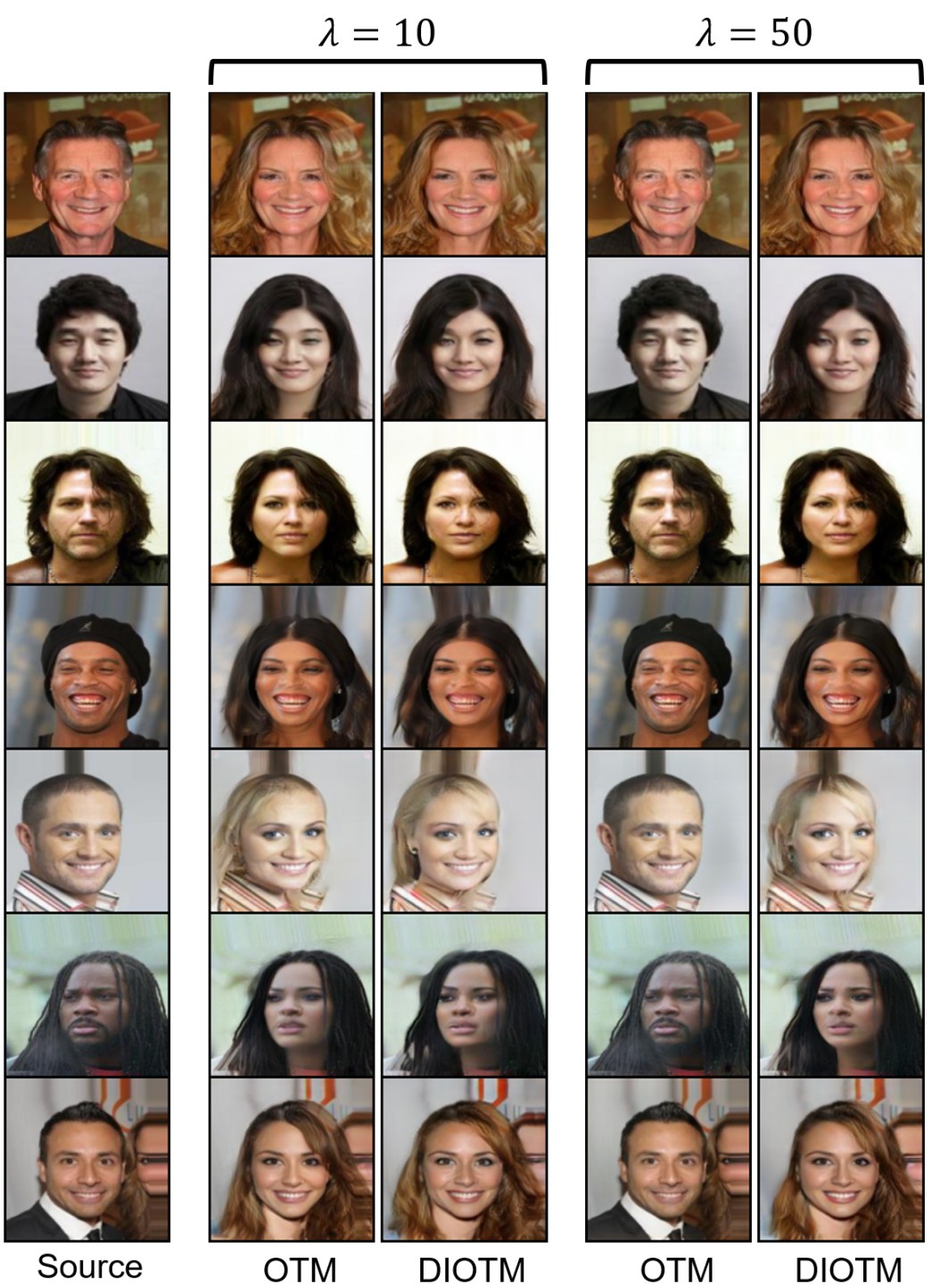

Figure 12: Qualitative comparison of unpaired Male → Female translation between OTM and DIOTM on 128 × 128 CelebA images.

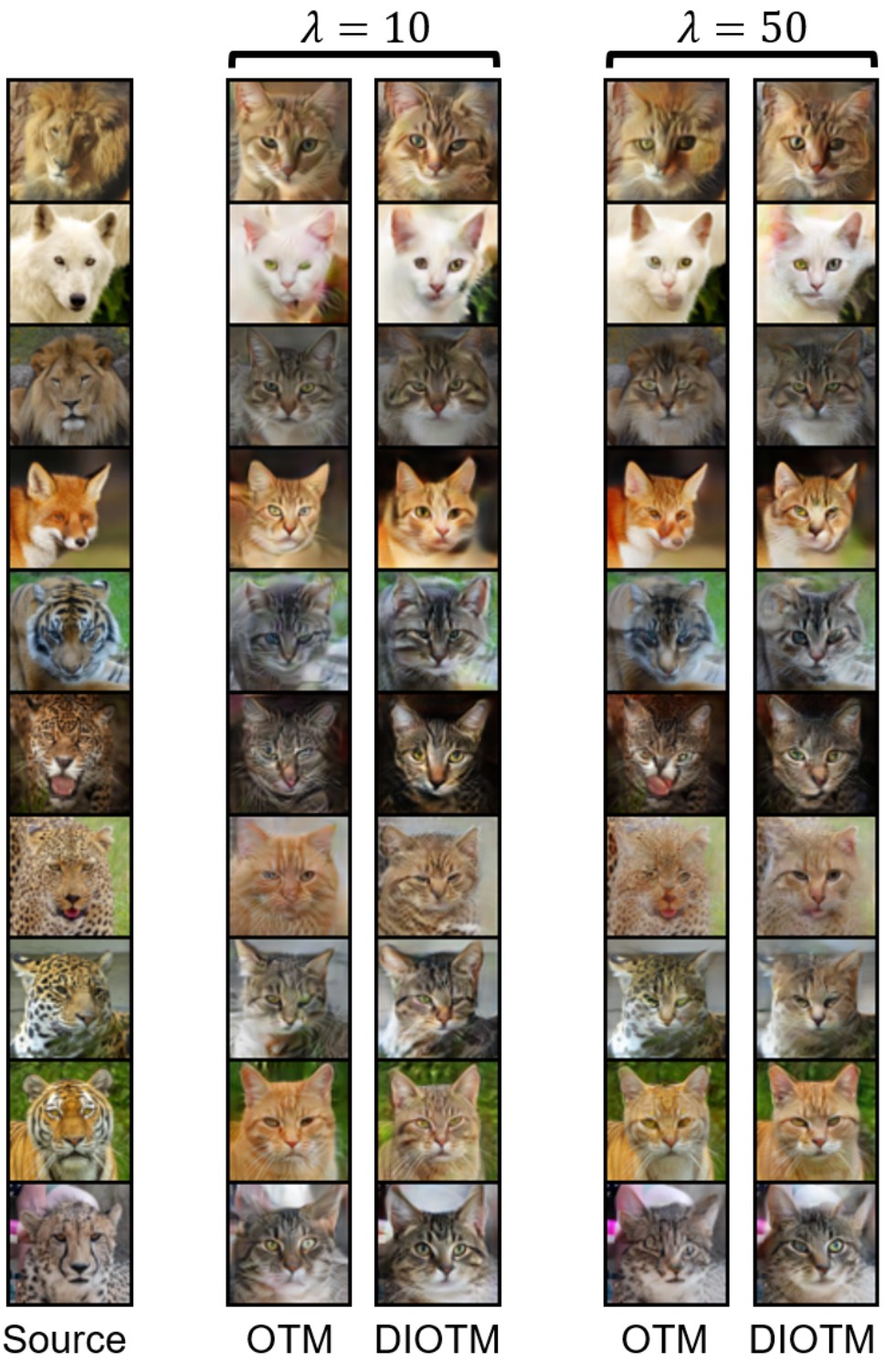

Figure 13: Qualitative comparison of unpaired Wild → Cat translation between OTM and DIOTM for 64 × 64 AFHQ images.

# D ADDITIONAL QUANTITATIVE RESULTS

## D.1 ADDITIONAL QUANTITATIVE RESULTS FOR IMAGE-TO-IMAGE TRANSLATION

**Evaluation of Image Alignment** To evaluate whether our transported image $T_\theta(x)$ preserves the properties of the original source image $x$, we compute the LPIPS (Zhang et al., 2018) metric between the source datapoint $x$ and its generation $T_\theta(x)$. The reported results are the mean LPIPS values calculated over a test dataset. As shown in Tab. 4, our model demonstrates comparable results to other image-to-image (I2I) benchmarks.

Table 4: Comparison of LPIPS ($\downarrow$) scores for perceptual similarity on Image-to-Image translation benchmarks.

| Model | Wild→Cat (64x64) | Male→Female (128x128) |
|-------|------------------|----------------------|
| DSBM  | 0.59 | 0.25 |
| OTM   | 0.47 | 0.21 |
| DIOTM | 0.45 | 0.25 |

**FID Curves along Training Process** As discussed in Sec. 5.3, our model demonstrates robustness to the regularization hyperparameter (Fig. 4) and maintains stable training dynamics (Fig. 5). In this paragraph, we provide the FID curves throughout the training process for various values of hyperparameter $\lambda$. Fig. 14 shows that our model exhibits stable convergence across various hyperparameters $\lambda$. In contrast, OTM converges only under specific hyperparameter settings, i.e., $\lambda = 10$ for Wild→Cat ($64 \times 64$) and $\lambda = 50$ for Male→Female ($128 \times 128$).

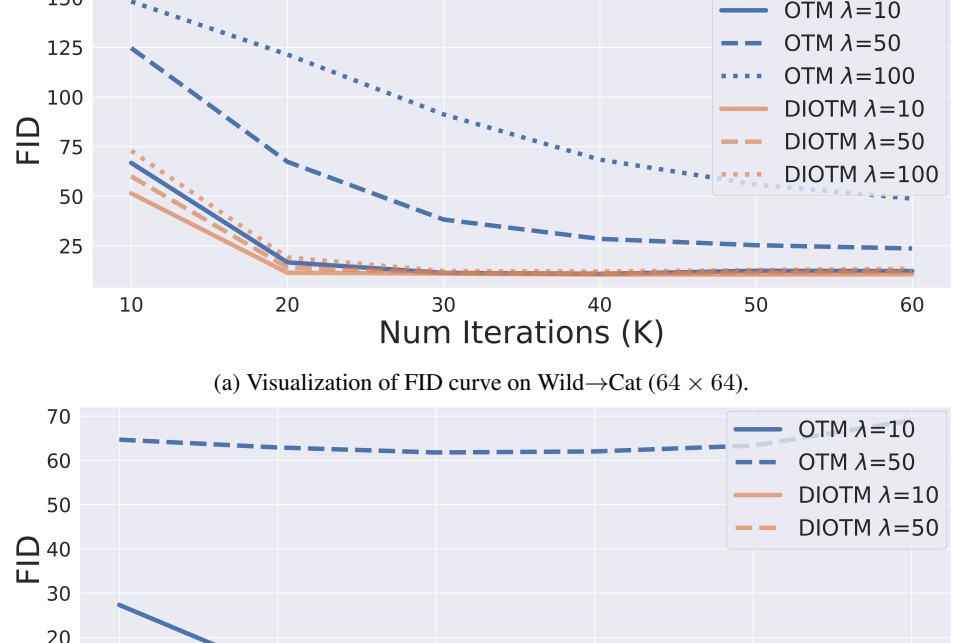

(a) Visualization of FID curve on Wild→Cat ($64 \times 64$).

(b) Visualization of FID curve on Male→Female ($128 \times 128$).

Figure 14: Visualization of FID ($\downarrow$) curves for various hyperparameter $\lambda$ on Image-to-Image translation tasks.

**Ablation Study on Time Sampling Distribution**  We investigate alternative time samplers for line 2 of Algorithm 1, beyond the uniform distribution $U[0,1]$. Specifically, we test two time sampling distributions: Beta$(2,2)$ and Beta$(0.5,0.5)$. The Beta$(2,2)$ distribution samples intermediate time values ($t \approx 0.5$) more frequently, while Beta$(0.5,0.5)$ tends to sample near the boundaries of the interval ($t \approx 0,1$) more often. This ablation study on time sampling distribution is evaluated on the Wild→Cat dataset. As shown in Table 5, the uniform sampler achieves slightly better performance compared to the other two sampling methods. However, the differences in performance are relatively minor, indicating that the alternative sampling methods yield results comparable to the uniform sampler.

Table 5: Ablation study on the time sampling distribution. FID scores are evaluated on the Wild→Cat ($64 \times 64$) dataset.

| Sampling Distribution | Uniform | Beta(0.5,0.5) | Beta(2,2) |
|---|---|---|---|
| FID ($\downarrow$) | **10.72** | 11.85 | 12.45 |

## D.2 Additional Quantitative Results for 2D-Toy Experiments

**Cyclical Property**  Our DIOTM model learns the bidirectional transport maps $\overrightarrow{T_\theta}, \overleftarrow{T_\theta}$ simultaneously. Here, $\overrightarrow{T_\theta}$ and $\overleftarrow{T_\theta}$ denote the optimal transport maps for $\mu \to \nu$ and $\nu \to \mu$, respectively. Theoretically, these two optimal transport maps satisfy $T_2 \circ T_1(x) = x$ and $T_1 \circ T_2(y) = y$ almost surely under our assumptions. We refer to this property as a *cyclical property*. To evaluate whether our model satisfies this cyclical property, we measured the MSE reconstruction error between $x$ and $T_2 \circ T_1(x)$, i.e. $\mathbb{E}_{x \sim \mu}[\|T_2 \circ T_1(x) - x\|^2]$. For comparison, we trained two OTM models for $\mu \to \nu$ and $\nu \to \mu$ and measured the MSE reconstruction error using these models.

As shown in Tab. 6, our DIOTM achieves better reconstruction error on four out of six experiments. Our model shows a larger reconstruction error in the 25Gaussian-to-Gaussian-to-25Gaussian case ($25G \to G \to 25G$). However, this reconstruction is meaningful when the generating distribution errors are also considered, i.e., $\overrightarrow{T_\theta}_\# \mu \approx \nu$ and $\overleftarrow{T_\theta}_\# \nu \approx \mu$ (Table 1). We interpret this result as being due to the larger distribution error of OTM in the $G \to 25G$ case.

Table 6: Evaluation of the cyclical property on synthetic datasets.

| Model | 8G→G→8G | G→8G→G | 25G→G→25G | G→25G→G | M→S→M | S→M→S |
|---|---|---|---|---|---|---|
| OTM | 1.06 | 0.040 | **4.13** | **0.56** | 4.01 | 1.12 |
| DIOTM | **0.22** | **0.015** | 12.43 | 0.68 | **1.11** | **0.46** |

## D.3 Comparison to OT Benchmarks on High Dimensional Data

Table 7: Continuous neural optimal transport benchmark (Korotin et al., 2021). The transport maps are evaluated based on the $\mathcal{L}^2$-UVP (%, $\downarrow$) metric and Cosine Similarities ($\uparrow$). The baseline scores are taken from Korotin et al. (2021) and † indicates the results by ourselves.

| Dimension | D=16 | | D=64 | |
|---|---|---|---|---|
| Metric | $\mathcal{L}^2$-UVP ($\downarrow$) | cos ($\uparrow$) | $\mathcal{L}^2$-UVP ($\downarrow$) | cos ($\uparrow$) |
| L | 41.6 | 0.73 | 63.9 | 0.75 |
| QC | 47.2 | 0.70 | 75.2 | 0.70 |
| MM | 2.2 | 0.99 | 3.2 | 0.99 |
| MMv1 | 1.4 | 0.99 | 8.1 | 0.97 |
| MMv2 | 3.1 | 0.99 | 10.1 | 0.96 |
| MMv2:R | 7.7 | 0.96 | 6.8 | 0.97 |
| MM-B | 6.4 | 0.96 | 13.9 | 0.94 |
| DIOTM† | 2.5 | 0.98 | 10.4 | 0.96 |

**High-dimensional Experiment**    In this section, we assess our model using a high-dimensional Gaussian mixture experiment, following the benchmark proposed in Korotin et al. (2021). Table 7 presents the results. Our model demonstrated reasonable performance, effectively learning the optimal transport map. Still, our model does not achieve state-of-the-art results. We believe this is because our primary goal was to achieve better scalability and stable estimation on a high-dimensional image dataset. To achieve this, our DIOTM does not impose specific structure constraints on two neural networks in our min-max optimization algorithm, i.e., the transport map and the discriminator, such as the input convex neural networks ICNN (Amos et al., 2017). Note that most of the baselines in Table 7 use ICNNs to parameterize their potential functions (discriminator). However, this approach sacrifices scalability, particularly for image datasets.

**Experimental Settings**    Unless otherwise stated, we follow the hyperparameter and the implementation of Korotin et al. (2021). We use the number of iterations of 100K, Adam optimizer of $(\beta_1, \beta_2) = (0, 0.9)$ with the learning rate of $10^{-3}$.

For both forward and backward transport maps, we use 3-MLP layers with SiLU activation function. For the discriminator, we embed time variable $t$ by using sinusoidal embedding of dimension 128, and then pass it through the two MLP layers with SiLU activation. For the state variable $x$, we embed it through 2-MLP layers. Then, we add time and state embeddings and pass them through a 3-layer MLP with SiLU activation. We use the hidden dimension of 2048 for all MLP layers.

