# OpenReview forum: "Improving Neural Optimal Transport via Displacement Interpolation"
_ICLR.cc/2025/Conference — ICLR 2025 Poster_

### Official Review · Reviewer_Zhh7 · 2024-11-02

**Soundness:** 3
**Presentation:** 3
**Contribution:** 3
**Rating:** 6
**Confidence:** 4

**Summary:**

This paper proposes a new method to estimate the optimal transport map between two distributions - a source and a target. The proposed method, called Displacement Interpolation Optimal Transport (DIOTM), leverages displacement interpolation which is the optimal solution of a particular dynamic formulation of OT with quadratic cost. The core component of the training algorithm for DIOTM involves a min-max loss objective, similar to GAN framework. This min-max objective is derived from the dual problem of the original minimization problem of displacement interplant. The expression involves a supremum over two potential value functions which can be combined into a single potential value function. These potential functions play a role similar to discriminators in GANs and the transport maps are similar to generators. In addition, the regularization term of the loss objective is derived from Hamilton-Jacobi-Bellman (HJB) optimality condition of the value function. The training algorithm involves alternately updating the potential value function and  the two transport maps. The paper shows applications of the proposed approach on image-to-image translation on datasets such as Male $\rightarrow$ Female (64$\times$64, 128$\times$128), Wild $\rightarrow$ Cat (64$\times$64), etc.

**Strengths:**

**Writing**: The paper is well-written. It provides sufficient background on major concepts involved in DIOTM such as displacement interpolation. The core algorithm has been explained well and the underlying theoretically motivation has been explained well.

**Quality and significance**: Improving stability of Optimal transport is an important problem and this paper proposes a method to addresses it.
- The experimental results on simple 2D toy datasets seem to indicate improved performance compared to prior methods as indicated in Table 1.
- DIOTM seems to outperform other optimal transport based models on image-to-image translation task in terms of metrics such as FID (Table 2).
- The proposed HJB regularizer seems to help with improved training dynamics (Figure 5). Further, HJB regularizer seems to be less sensitivity to the choice of regularization hyperparameter  (Table 3) which is a desirable property.

**Weaknesses:**

1. This method trains two optimal transport maps from source to target distribution and vice versa which is a bit inefficient. Further, there are no experiments which demonstrate that the two independently trained transport maps are invertible, which they should be theoretically. How does source -> target -> source reconstruction perform on various datasets in the paper? Similarly, target -> source -> target reconstruction on images should be reported with a metric such as l2 error/reconstruction error.
2. Qualitative Results: The paper should Include qualitative comparison with other methods on Image-to-Image translation baseline. FID doesn’t necessarily capture lot of semantic and perceptual information of images. A better comparison would be side-by-side comparison of images obtained from DIOTM and previous OT benchmarks.
3. Quantitative results: Table 2 compares DIOTM with existing neural optimal transport models. For the sake of completeness, the paper should include another table that includes other state-of-the-art methods (e.g. GANs[1], flows as well as diffusion-based methods (e.g. Wang et al. [2]) for image-to-image translation task so that reader gets an overall picture of the landscape and the gap of DIOTM from SOTA method. I would like to reiterate that it is completely alright if DIOTM is not SOTA overall, compared to other methods for I2I task, but such a table should be included, as it is a standard practice.
4. Implementation details: The paper is missing some of the implementation details, specifically architecture details of networks for image-to-image  translation task. Further, the number of images used to calculate FID is unclear.
5. The largest image resolution considered in this work is 128X128 which is not very large. In order to reliably evaluate scalability, larger resolutions such as 256X256 or 512X512 should be considered. See Isola et al. [1] for a list of potential datasets for image-to-image translation tasks on larger resolution.

[1] Isola, Phillip, et al. "Image-to-image translation with conditional adversarial networks." Proceedings of the IEEE conference on computer vision and pattern recognition. 2017.
[2] Wang, Tengfei, et al. "Pretraining is all you need for image-to-image translation." arXiv preprint arXiv:2205.12952 (2022).

**Questions:**

1. Training stability: Can we have multiple curves to understand how frequently the training diverges for OTM? Also, how sensitive is training of OTM to various hyperparameters?
2. What are some practical constraints on the source and target distributions when trying to learn an OT map with DIOTM? Can it learn OT map in the cases where the distance between the source and target distribution might be large? For instance, prior works in this space consider more complex datasets/tasks for image-to-image translation such as mask-to-image synthesis (COCO / ADE-20K), sketch-to-image synthesis, day-to-night, summer-to-winter, colorization etc.
3. The results of Figure 11 seem much more suboptimal than other cases (with multiple faces) etc. What could be the reason for more failures for this  pair of distribution?
Minor:
- Line 243: typo - parametrization
- Line 253, 256, 258: Consider using different parameter notation e.g. $\overrightarrow{T}_\theta$ and $\overleftarrow{T}_\tilde{\theta}$ for the two transport maps, as these are parametrized with two different networks with different parameters. This would make it clear that these two networks are trained separately, as opposed to using a shared network.
- Repeated citation for Diffusion Schrodinger bridge matching paper.

---

> ### Author Response · Authors · 2024-11-22
> **Response to Reviewer Zhh7 (1/2)**
>
> We sincerely thank the reviewer for carefully reading our manuscript and providing valuable feedback. We appreciate the reviewer for considering our work "well-written" and for acknowledging that "improving the stability of Optimal transport is an important problem". We hope our responses to be helpful in addressing the reviewer’s concerns. We highlighted the corresponding revisions in the manuscript in Brown.
>
> $ $
>
> ---
>
> **W1.**
> (1)This method trains two optimal transport maps from source to target distribution and vice versa which is a bit inefficient.
>
> (2) Further, there are no experiments which demonstrate that the two independently trained transport maps are invertible, which they should be theoretically. How does source -> target -> source reconstruction perform on various datasets in the paper? Similarly, target -> source -> target reconstruction on images should be reported with a metric such as l2 error/reconstruction error.
>
> **A.**
> (1) As discussed in the limitation part of the Conclusion section (Lines 526-530), we agree with the reviewer that our DIOTM is less efficient than OTM due to the requirement to train two optimal transport maps. Nevertheless, **our approach is significantly more efficient than dynamical OT methods, such as DSBM.** By leveraging the displacement interpolation parametrization (Eq. 17), our model eliminates the need for intermediate time simulations and offers efficient generation with only 1 NFE (Number of Function Evaluations).
>
>
> (2) **We evaluated the reconstruction error of DIOTM using its bidirectional transport maps**. As a baseline, we trained two OTM models independently in both directions, i.e., $\mu \rightarrow \nu$ and $\nu \rightarrow \mu$, and measured its reconstruction error. The results are presented below:
>
> - Reconstruction error comparison of DIOTM and OTM
>
> |Model|8G $\rightarrow$ G $\rightarrow$ 8G|G $\rightarrow$ 8G $\rightarrow$ G|25G $\rightarrow$ G $\rightarrow$ 25G|G $\rightarrow$ 25G $\rightarrow$ G|M $\rightarrow$ S $\rightarrow$ M|S $\rightarrow$ M $\rightarrow$ S|
> |:---|:---|:---|:---|:---|:---|:---|
> |OTM|   1.06    | 0.040 |  **4.13**  |  **0.56**  | 4.01  | 1.12  |
> |DIOTM| **0.22**    | **0.015** | 12.43  |  0.68  | **1.11**  | **0.46**  |
>
> Our DIOTM achieves better reconstruction error on four out of six experiments. Our model shows a larger reconstruction error in the 25Gaussian-to-Gaussian-to-25Gaussian case ($25G \rightarrow G \rightarrow 25G$). However, this reconstruction is meaningful when the generating distribution errors are also considered, i.e., $\overrightarrow{T\_{\theta}}\_{\\#} \mu \approx \nu$ and $\overleftarrow{T\_{\theta}}\_{\\#} \nu \approx \mu$ (Table 1). We interpret this result as being due to the larger distribution error of OTM in the $G \rightarrow 25G$ case.
>
> $ $
>
> ---
>
> **W2.**
> Qualitative Results: The paper should Include qualitative comparison with other methods on Image-to-Image translation baseline. FID doesn’t necessarily capture lot of semantic and perceptual information of images. A better comparison would be side-by-side comparison of images obtained from DIOTM and previous OT benchmarks.
>
> **A.**
> We included **additional qualitative comparisons of translated samples between OTM and DIOTM** in Fig 12 (Male $\rightarrow$ Female 128x128) and Fig 13 (Wiid $\rightarrow$ Cat 64x64) in the Appendix. These examples demonstrate that OTM often fails to transform target semantics correctly, e.g. resulting in Male $\rightarrow$ Male in $\lambda=50$. Furthermore, we evaluated the LPIPS score ($\downarrow)$ to quantitatively evaluate how well the generator preserves the content. The results are presented in Table 4 in the Appendix. Our model exhibits comparable LPIPS scores to the baselines.
>
> $ $
>
> ---
>
> **W3.**
> Quantitative results: Table 2 compares DIOTM with existing neural optimal transport models. For the sake of completeness, the paper should include another table that includes other state-of-the-art methods (e.g. GANs[1], flows as well as diffusion-based methods (e.g. Wang et al. [2]) for image-to-image translation task so that reader gets an overall picture of the landscape and the gap of DIOTM from SOTA method. I would like to reiterate that it is completely alright if DIOTM is not SOTA overall, compared to other methods for I2I task, but such a table should be included, as it is a standard practice.
>
> **A.**
> In this work, our scope was to compare with OT map approaches for unpaired image-to-image translation tasks. However, **we respectfully believe that these two works are supervised approaches, and therefore not directly comparable to our unsupervised model**. Specifically, [1] employs a conditional GAN and [2] utilizes large-scale pretraining with a diffusion decoder, using the paired data. As our model is designed for unpaired image-to-image translation tasks, these methods are not appropriate for direct comparison.

---

> ### Author Response · Authors · 2024-11-22
> **Response to Reviewer Zhh7 (2/2)**
>
> ---
>
> **W4.**
> Implementation details: The paper is missing some of the implementation details, specifically architecture details of networks for image-to-image translation task. Further, the number of images used to calculate FID is unclear.
>
> **A.**
> As detailed in Appendix B.2, **we utilized the NCSN++ [1] as the backbone network**. Note that all methods (DSBM, ASBM, OTM, and Ours) in the Image-to-Image translation tasks (Table 2) used the same backbone network. Also, regarding the FID computation, we followed the **evaluation scheme of [1] for the Wild→Cat experiments and [2] for the CelebA experiments for a fair comparison**. Specifically, in the Wild→Cat experiments, we generated ten samples for each source test image. Since the source test dataset consists of approximately 500 samples, we generated 5000 generated samples. Then, we computed the FID score with the training target dataset, which also contains 5000 samples. Also, in the CelebA experiment, we computed the FID score using the test target dataset, which includes 12247 samples. We generated the same 12247 samples and compared them with the test target dataset. We revised this part in the Appendix B.2 for better clarity.
>
> $ $
>
> [1] De Bortoli, Valentin, et al. "Schr\" odinger Bridge Flow for Unpaired Data Translation." NeurIPS 2024.
> [2] Gushchin, Nikita, et al. "Adversarial Schr\" odinger Bridge Matching." NeurIPS 2024.
>
> $ $
>
> ---
>
> **W5.**
> The largest image resolution considered in this work is 128X128 which is not very large. In order to reliably evaluate scalability, larger resolutions such as 256X256 or 512X512 should be considered. See Isola et al. [1] for a list of potential datasets for image-to-image translation tasks on larger resolution.
>
> **A.**
> Due to computational resource constraints, we were unable to conduct experiments on larger resolutions such as 256X256 or 512X512. However, considering that existing approaches in the OT literature usually evaluate their methods on resolutions comparable to or lower than 128x128 [1,2,3],
> we believe that using 128x128 resolution images can also be considered a high-dimensional experiment in the OT literature.
>
> $ $
>
> [1] Fan, Jiaojiao, et al. "Scalable computation of monge maps with general costs." Arxiv.
> [2] Rout, Litu, Alexander Korotin, and Evgeny Burnaev. "Generative modeling with optimal transport maps." ICLR 2022.
> [3] Gushchin, Nikita, et al. "Adversarial Schr\" odinger Bridge Matching." NeurIPS 2024.
>
> $ $
>
> ---
>
> **Q1.**
> Training stability: Can we have multiple curves to understand how frequently the training diverges for OTM? Also, how sensitive is training of OTM to various hyperparameters?
>
> **A.**
> As shown in Fig. 4, the perfomance of OTM is highly sensitive to the regularization hyperparameter $\lambda$. In contrast, our model demonstrates robust performance to $\lambda$.
>
> $ $
>
> ---
>
> **Q2.**
> What are some practical constraints on the source and target distributions when trying to learn an OT map with DIOTM? Can it learn OT map in the cases where the distance between the source and target distribution might be large? For instance, prior works in this space consider more complex datasets/tasks for image-to-image translation such as mask-to-image synthesis (COCO / ADE-20K), sketch-to-image synthesis, day-to-night, summer-to-winter, colorization etc.
>
> **A.**
> We appreciate the reviewer for the thoughtful comment. The primary limitation of DIOTM for more complex tasks is that **our model assumes a quadratic cost function** $c(x,y) = \alpha \| x - y \|^{2}$. This is because the equivalence between displacement interpolation and dynamic optimal transport (Eq. 7) does not hold for general cost functions. As a result, DIOTM is not applicable to tasks where the pixel-wise quadratic cost is inappropriate. In this work, we focused on improving the stability of neural optimal transport for relatively low-resolution data. However, we believe generalizing neural optimal transport approaches to more challenging high-resolution data would be an important direction for future research.
>
> $ $
>
> ---
>
> **Q3.**
> The results of Figure 11 seem much more suboptimal than other cases (with multiple faces) etc. What could be the reason for more failures for this pair of distribution?
>
> **A.**
> We believe that this result happens because of **the characteristics of the Wild and Cat distributions**. In Wild images, black animals are relatively rare, except for tigers with black stripes (Fig 10, Left).  Consequently, black cats are usually translated into the tiger with black stripes (Fig 11). However, this translation incurs a relatively high cost, compared to the Male $\leftrightarrow$ Female case. We think that these unbalanced source and target distributions make image-to-image translation via the Optimal transport map more challenging in the Wild $\leftrightarrow$ Cat datasets.
>
> $ $
>
> ---
>
> **Minors/Typos**
>
> **A.**
> Thank you for the careful advice. We corrected the manuscript accordingly.

---

> > ### Comment · Reviewer_Zhh7 · 2024-11-27
> >
> > Thank you for your replies. My major concerns have been adequately addressed. After going through the revision, I think this paper should be accepted.

---

> ### Author Response · Authors · 2024-11-27
>
> We are glad to know we have addressed most of your comments. Thanks again for your feedback and appreciation of our work. We are happy to take additional questions or suggestions before the rebuttal period ends.

---

### Official Review · Reviewer_WWpL · 2024-11-05

**Soundness:** 2
**Presentation:** 2
**Contribution:** 2
**Rating:** 6
**Confidence:** 3

**Summary:**

This paper builds on displacement interpolation in Optimal Transport (OT) and introduces a time-derivative HJB regularizer, enhancing training stability. The training of the model is based on min-max optimization similar to GAN. It achieves state-of-the-art results on both synthetic data and image-to-image translation tasks w.r.t $W_2$, $L_2$ and FID score.

**Strengths:**

1. The paper presents comprehensive and detailed theoretical derivations, with notable innovations within the OT framework. It leverages the dual formulation of displacement interpolation to derive a new min-max optimization function.

2. In terms of experimental performance, the proposed HJB regularizer is effectively insensitive to the hyperparameter $\lambda$, performing better than other regularizers such as R1 and OTM. And DIOTM outperforms other benchmarks and exhibits more stable training.

**Weaknesses:**

1. The motivation behind the theoretical innovation is unclear. There is no analysis explaining why decomposing the optimization of $T_\theta$ in OTM into optimizations for forward $\overrightarrow{T_\theta}$ and backward $\overleftarrow{T_\theta}$ improves training stability.

2. The experimental results in Table 2 appear unusual. I couldn't find related experimental setups for the benchmarks, and some references don’t report similar experiments or use different resolution datasets. Since the FID scores for these benchmarks couldn’t be directly cited, how were these results obtained? Were all models trained for the same number of steps? It would be beneficial to add an ablation study of FID vs. training steps.

3. The paper argues that DIOTM is more stable than OTM, but Fig. 5 shows that OTM remains stable for the first 40K steps before experiencing a sudden spike in loss. What caused this increase? If the loss curve does not decrease further, why train for 60K steps rather than 40K?

4. The paper only provides visualizations for DIOTM, making it hard to compare visually with baselines. The DSBM paper’s wild-to-cat results at 512x512 resolution look much better than those in Fig. 2, yet its FID score in Table 2 is much higher. Could the authors clarify this discrepancy?

**Questions:**

Refer to weaknesses.

---

> ### Author Response · Authors · 2024-11-22
> **Response to Reviewer WWpL (1/2)**
>
> We sincerely thank the reviewer for carefully reading our manuscript and providing valuable feedback. We are delighted that the reviewer considers our work to present "notable innovations within the OT framework". We hope our responses to be helpful in addressing the reviewer's concerns. We highlighted the corresponding revisions in the manuscript in Red.
>
> $ $
>
> ---
>
> **W1.** The motivation behind the theoretical innovation is unclear. There is no analysis explaining why decomposing the optimization of $T_{\theta}$ in OTM into optimizations for forward $\overrightarrow{T}\_{\theta}$ and backward $\overleftarrow{T}\_{\theta}$ improves training stability.
>
> **A.**
> Our intuitive motivation is that our DIOTM can achieve more stable training by matching the entire intermediate dynamics of the probability distribution, from the source distribution $\mu$ to the target distribution $\nu$.
>
> - The neural optimal transport estimation becomes more challenging when the source and target distributions are far apart, such as in high-dimensional datasets. In DIOTM, the forward and backward transport maps $\overrightarrow{T}, \overleftarrow{T}$ are trained to match all intermediate distributions. As a result, **each transport map is not directly trained to generate the boundary distributions $\mu$ and $\nu$, but instead exploits the matching of intermediate distributions**. This approach enables our model to achieve a more stable estimation of the optimal transport map.
> - In practice, a common challenge in adversarial training is that the discriminator learns faster than the generator. In this regard, our scheme addresses this by imposing a more challenging task on the value function (discriminator) $V(t, x)$, because $V(t, x)$ is trained to discriminate samples across all intermediate time $t$. Therefore, **our scheme achieves a better balance between the transport map and the value function**.
>
> We incorporated this intuition in Lines 240-244 in the revised version of our manuscript.
>
> $ $
>
> ---
>
> **W2.**
> The experimental results in Table 2 appear unusual. I couldn't find related experimental setups for the benchmarks, and some references don’t report similar experiments or use different resolution datasets. Since the FID scores for these benchmarks couldn’t be directly cited, how were these results obtained? Were all models trained for the same number of steps? It would be beneficial to add an ablation study of FID vs. training steps.
>
> **A.**
> **The FID values for DSBM in Table 2 are taken from [1] for Wild $\rightarrow$ Cat (64x64) and [2] for Male $\rightarrow$ female (128x128)**. All other FID values without the $\dagger$ symbol are taken from their original papers. In the revised version of our manuscript, we marked the results conducted by us with a $\dagger$ and revised the caption of Table 2 accordingly.
>
> Moreover, we included the FID vs. Training iterations plot in Fig 14 of the Appendix. This plot shows that our DIOTM achieves more stable FID dynamics across diverse regularization parameter $\lambda$ values.
>
> $ $
>
> [1] De Bortoli, V., et al. "Schr\"{o}dinger Bridge Flow for Unpaired Data Translation." NeurIPS 2024.
> [2] Gushchin, N., et al. "Adversarial Schr\"{o}dinger Bridge Matching." NeurIPS 2024.

---

> > ### Comment · Reviewer_WWpL · 2024-11-22
> >
> > Thanks for the reply to my questions about the motivation for optimization design and the experimental results. I see the robustness of DIOTM w.r.t. the regularization parameter $\lambda$. I raise my rating to 6 in this case.
> >
> > However, the OTM seems able to achieve a similar performance compared with DIOTM under a good $\lambda$ (Fig. 14).

---

> ### Author Response · Authors · 2024-11-22
> **Response to Reviewer WWpL (2/2)**
>
> $ $
>
> ---
>
> **W3.** The paper argues that DIOTM is more stable than OTM, but Fig. 5 shows that OTM remains stable for the first 40K steps before experiencing a sudden spike in loss. What caused this increase? If the loss curve does not decrease further, why train for 60K steps rather than 40K?
>
> **A.** In Fig 5, we observed a sudden spike in training loss for OTM, consistent with the behavior reported in [1]. We hypothesize this is due to the balancing problem between the generator and discriminator in adversarial training, as discussed in our response to W1.
>
> Regarding the train iterations, first, we would like to emphasize that **our model is trained using a min-max learning objective (Eq. 19)**. Therefore, the loss minimization cannot be interpreted as a sign of model training. Moreover, in practice, the loss does not decrease after a certain level, as we observe in Fig 5. Hence, we cannot determine the optimal training iterations based on the loss value.
>
> Second, as the reviewer suggested, we can alternatively train OTM for 40K iterations instead of 60K iterations. However, we believe that **identifying the optimal stopping point during training without access to the test dataset is still a problem in practice**. Hence, we believe this sudden training instability remains a fundamental issue with OTM. Furthermore, in addition to the training loss plot in Fig 5, we would like to emphasize that Fig 4 shows that our model is more **robust to the regularizer hyperparameter** $\lambda$ than OTM. Note that we reported the best FID score for OTM from each 10K training iteration. Our DIOTM achieves superior results, even when compared to the best-case result of OTM before collapsing.
>
> $ $
>
> [1] Choi, J., Choi, J., and Kang, M. "Analyzing and Improving Optimal-Transport-based Adversarial Networks." ICLR 2024.
>
> $ $
>
> ---
>
> **W4.**
> The paper only provides visualizations for DIOTM, making it hard to compare visually with baselines. The DSBM paper’s wild-to-cat results at 512x512 resolution look much better than those in Fig. 2, yet its FID score in Table 2 is much higher. Could the authors clarify this discrepancy?
>
> **A.**
> Please refer to our response in W2.

---

> > ### Comment · Reviewer_WWpL · 2024-11-23
> >
> > As shown in Fig. 14, 40K training is good enough regarding FID score. I think an early stopping strategy is easy to implement, given the change in loss (see Fig.5). So the authors' response to W3 doesn't convince me.

---

> ### Author Response · Authors · 2024-11-25
>
> Thank you for reviewing our paper! We agree with the reviewer that OTM demonstrates competitive performance with the best hyperparameter $\lambda$. As shown in Table 2, we would like to remind that DIOTM achieves a better FID score compared to the best FID score of OTM. We appreciate the reviewer's valuable feedback.

---

### Official Review · Reviewer_uCi4 · 2024-11-08

**Soundness:** 3
**Presentation:** 3
**Contribution:** 3
**Rating:** 8
**Confidence:** 3

**Summary:**

The paper presents a theoretically justified method for the computation of dynamic optimal transport using the theory of Displacement Interpolation. The authors derive a dual formulation of Displacement Interpolation. They show that the optimal potential for solving the dual problem satisfies the HJB equation and incorporates the HJB equation as a regularizer for the training of the proposed method. The method is validated on synthetic datasets - G->8G, G->25G, Moon->Spiral, G->Circles - and several image-to-image translation problems - Celeba and Wild->Cat. The proposed method achieves the best FID among considered competitors CycleGAN, OTM, NOT, DSBM, and ASBM for image-to-image translation problems and outperforms closely related OTM method on most synthetic datasets in W^2 distance.

**Strengths:**

1. The method has a derivation of the dual problem for displacement interpolation, which opens the possibility of numerical optimal transport computation from the perspective of the Benamou-Brenier dynamic transport formulation.
2. Experiments on toy examples and image-to-image translation problems show that the proposed method achieves good numerical results over competing methods for optimal transport computation and is scalable to image problems.
3. The paper proves numerically that the HJB regularizer improves the training procedure and is better than the OTM and R1 regularizers. This regularizer seems to be novel in the literature of numerical optimal transport computation.

**Weaknesses:**

1. The method doesn't compare to closely related flow-based optimal transport methods, such as Rectified Flow (Flow straight and fast: Learning to generate and transfer data with rectified flow, ICLR-2023) and Flow Matching (Flow Matching for Generative Modeling, ICLR-2023). I suggest the authors compare with these methods as well.
2. The paper lacks a visual comparison for image-to-image translation problems between different methods and a discussion of why competing methods perform worse. It is not clear why the proposed method achieves better numerical results when it has similar visual results to competitors.
3. It is not clear how well the method computes optimal transport in high dimensions. I suggest that the authors evaluate their method on the Wasserstein-2 benchmark (Do neural optimal transport solvers work? A continuous Wasserstein-2 benchmark, NeurIPS-2021).

**Questions:**

Questions:
1. It looks like in Eq. 6 the integration should be over $x$ instead of $d\rho_ {t}$, and $\rho_ {t}(x)$ should be under the integral. Can you comment on this?
2. Can you clarify how long it took you to train your methods for image-to-image translation problems compared to competing methods?
3. What is the number of parameters used by all the methods for image-to-image translation problems? Are they comparable?
4. Have you experienced failures of your method, and if so, can you provide them?
5. Can you provide an evaluation of your method on the Wasserstein-2 benchmark to show that the method is capable of solving optimal transport in high dimensions?
Typing errors:
1. Line 49 - double "the"; one should be deleted.

---

> ### Author Response · Authors · 2024-11-22
> **Response to Reviewer uCi4 (1/2)**
>
> We sincerely thank the reviewer for carefully reading our manuscript and providing valuable feedback. We are delighted that the reviewer considers this work "opens the possibility of neural OT from the perspective of the dynamic OT" and finds our HJB regularizer novel. We hope our responses to be helpful in addressing the reviewer's concerns.
>
> $ $
>
> ---
>
> **W1.**
> The method doesn't compare to closely related flow-based optimal transport methods, such as Rectified Flow (Flow straight and fast: Learning to generate and transfer data with rectified flow, ICLR-2023) and Flow Matching (Flow Matching for Generative Modeling, ICLR-2023). I suggest the authors compare with these methods as well.
>
> **A.**
> We respectfully believe that Rectified Flow and Flow Matching are beyond the scope of our work. In this work, our scope was to compare with exact approaches for learning OT maps, particularly those targeting image-to-image translation tasks. Flow Matching learns the marginal velocity field connecting two distributions. While Flow Matching utilizes OT for its conditional probability path, the marginal velocity field does not correspond to the optimal transport map ($v_{t}$ in Eq 6). Similarly, Rectified Flow is an asymptotic process for refining Flow Matching into the optimal transport map, but it is not directly itself the optimal transport map. However, we agree with the reviewer that generalizing our approach to the dynamic optimal transport problem would be a promising direction for future research. Also, we agree that these works are broadly related to OT problems. Hence, we cited these works in Line 36 of our manuscript.
>
> $ $
>
> ---
>
> **W2.**
> The paper lacks a visual comparison for image-to-image translation problems between different methods and a discussion of why competing methods perform worse. It is not clear why the proposed method achieves better numerical results when it has similar visual results to competitors.
>
> **A.**
> We included **additional qualitative comparisons of translated samples between OTM and DIOTM** in Fig 12 (Male $\rightarrow$ Female 128x128) and Fig 13 (Wiid $\rightarrow$ Cat 64x64) in the Appendix. These examples demonstrate that OTM often fails to transform target semantics correctly, e.g. resulting in Male $\rightarrow$ Male in $\lambda=50$.
>
> $ $
>
> ---
>
> **W3.**
> It is not clear how well the method computes optimal transport in high dimensions. I suggest that the authors evaluate their method on the Wasserstein-2 benchmark (Do neural optimal transport solvers work? A continuous Wasserstein-2 benchmark, NeurIPS-2021).
>
> **Q5.**
> Can you provide an evaluation of your method on the Wasserstein-2 benchmark to show that the method is capable of solving optimal transport in high dimensions?
>
> **A.**
> We would like to gently remind that **our DIOTM is also evaluated on the image-to-image translation task on 128x128 resolution images** (Table 2). Given that existing approaches in the OT literature usually evaluate their methods on comparable or lower resolution images [1,2,3], the 128x128 resolution images can be considered a high-dimensional experiment.
>
>
> Furthermore, following the reviewer's advice, we **conducted a high-dimensional evaluation on the Gaussian mixture benchmark**. The experimental results are in Table 6 in the Appendix. Our model demonstrated reasonable performance, effectively learning the optimal transport map. Still, our model does not achieve state-of-the-art results. We believe this is because our primary goal was to achieve better scalability and stable estimation on a high-dimensional image dataset. To achieve this, our DIOTM does not impose specific structure constraints on two neural networks, parametrizing the transport map and the discriminator, such as ICNN. However, such an approach (ICNN) sacrifices scalability, particularly for image datasets.
>
> $ $
>
> ---
>
> **Q1.**
> It looks like in Eq. 6 the integration should be over $x$ instead of $d \rho_{t}$, and $\rho_{t}(x)$ should be under the integral. Can you comment on this?
>
> **A.**
> Thank you for the careful comment. We revised Eq. 6 as the reviewer commented.
>
> $ $
>
> ---
>
> **Q2.**
> Can you clarify how long it took you to train your methods for image-to-image translation problems compared to competing methods?
>
> **A.**
> We compared the training time of our method with several previous works in Table 2. Because DIOTM trains transport maps in both directions, DIOTM requires more training time than OTM, which trains the transport map in one direction. However, our model trains significantly faster than dynamical approaches such as DSBM and ASBM.
>
> - Training time comparison for the Male$\rightarrow$Female (128x128) experiment
>
> |Model|OTM|DSBM/ASBM|Ours|
> |:---|:---|:---|:---|
> |Time| 15h| > 10days| 28h|

---

> > ### Author Response · Authors · 2024-11-22
> > **Response to Reviewer uCi4 (2/2)**
> >
> > ---
> >
> > **Q3.**
> > What is the number of parameters used by all the methods for image-to-image translation problems? Are they comparable?
> >
> > **A.**
> > All methods (DSBM, ASBM, OTM, and Ours) in the Image-to-Image translation tasks (Table 2) used the same NCSN++ [1] as the backbone network.
> >
> > [1] Song, Yang, et al. "Score-based generative modeling through stochastic differential equations." ICLR 2021.
> >
> > $ $
> >
> > ---
> >
> > **Q4.**
> > Have you experienced failures of your method, and if so, can you provide them?
> >
> > **A.**
> > Our method failed without any regularization (HJB, OTM, R1 in Table 3), similar to the previous approach (OTM). To address this, we established the relationship between the value functions for each displacement interpolation $\rho_{t}$ in Thm 3.3. Using this optimality condition (Eq. 16), we introduced our HJB regularizer.
> >
> > $ $
> >
> > ---
> >
> > **Typing errors:**
> > Line 49 - double "the"; one should be deleted.
> >
> > **A.**
> > Thank you for the careful advice. We corrected the manuscript accordingly.

---

### Official Review · Reviewer_xVqq · 2024-11-09

**Soundness:** 3
**Presentation:** 3
**Contribution:** 3
**Rating:** 6
**Confidence:** 4

**Summary:**

The authors propose a novel method (DIOTM) to solve the optimal transport mapping problem for the quadratic transport cost (Wasserstein-2 OT) with neural networks. The approach is ideologically inspired by the previous works in the field which solve the dual (semi-dual) optimal transport problem by approximating an OT map and the dual potential (a.k.a. discriminator) with neural networks and optimizing them in the GAN-style adversarial manner (max-min).

The key innovative thing in the current paper lies in exploiting the properties of the W2 OT maps. They are related to the displacement interpolation linear interpolation from the input distribution to the target using the OT map). More precisely, the authors formulate the (semi-)dual problem for finding the displacement interpolation for a given time moment t in (0,1) which requires optimizing a particular t-dependent dual potential. Then they group all these problems together and obtain a dual problem when they have to optimize over one t-conditioned dual potential (and also additional t-dependent transport maps). In principle, each problem for different t can be viewed as independent, but

1) The authors note that after some reparameterization, the t-dependent dual potentials should satisfy the Hamilton-Jacobi-Bellman (HJB) condition. At this point, the authors propose to incorporate the HJB-inspired regularization into the optimization, which helps connect optimization problems for each t together.

2) The authors note that the optimal transport maps at each time moment t are connected with each other. In fact, they all can be expressed through each other and through the main transport map (from source to target). As a result, the authors use restricted parameterization where all these t-dependent maps are parameterized through a single map.

The resulting algorithm is a (simulation free) bi-directional max-min adversarial training scheme. The authors demonstrate the superiority of the proposed technique compared to previous dual form neural optimal transport solvers & their regularization techniques through a series of experiments (toy 2D data + image-to-image translation).

**Strengths:**

1) The idea of exploiting the displacement interpolation overall looks interesting and fresh. To my knowledge, it has not been actively studied in the field, so I believe that further developing it may be interesting and fruitful for the community of adversarial/dual-based OT methods. Overall, the contribution of this paper looks as significant for the neural OT field, as WGAN-GP improved WGAN.

2) The HJB based-regularization proposed here seems to be very natural and unbiased in the sense that it looks theoretically justified and does not bias the resulting solution. This is not the case for other GAN-based regularizing techniques which appear in related works (like R1 or other gradient penalty regularizers). However, for me it is still not clear from the main text if the authors in their method use only HJB or HJB+R1. This should be clarified.

3) The experimental comparison on unpaired Image-2-Image looks rather convincing and supports the main claim that HJB regularizer is useful for stability and works (I deduce this from the results of comparison with various dual OT methods).

4) The text is overall readable and the clarity is ok (although sometimes the amount of the bolded text is too annoying).

**Weaknesses:**

1) I believe that there might be a theoretical gap in the proposed DI-OTM approach which lies in the restricted parameterization of the t-dependent transport maps. Specifically, each transport map (for a particular t) should be parameterized the way that it should solve the corresponding inner conjugation (c-transform) minimization for a particular corresponding dual potential (for time t). However, when the authors tighten all the transport maps together via a single function, this may not hold and may spoil the theoretical validity of the proposed semi-dual form. This aspect should be discussed in more detail.

2) I think that some of the results presented here are not completely novel and the authors miss a large set of related work. The key problem which is exploited in the current work is the displacement interpolation optimization (equation 8). In essence, this is the Wasserstein-2 barycenter problem and, to my understanding, it has already been well studied both in theory and in practice. For example, the W2 dual barycenter problem (equation 9 in theorem 3.1 in the current paper) has been derived in the founding work [1], see their derivations around proposition 2.2. The semi-dual version (which is the second part of theorem 3.1 in the current paper) seems to directly follow from the general semi-dual for barycenters which has been recently introduced in [2] (theorem 4.1). I think these relations to the barycenter literature (theoretical and computational) should be clearly clarified and the related literature should be included.

3) The DIOTM approach proposed here seems to work only for the quadratic cost optimal transport (and may be for some lp-based OT as well) due to reliance on the displacement interpolation properties. It looks like it can not be generalized to more general OT formulations, e.g., formulations with non-lp transport costs. This point is more a limitation than a weakness as the authors specifically target the quadratic cost OT. Nevertheless, it should be mentioned in the paper and the background considers the general cost OT.

4) While the authors claim that they significantly improve the accuracy of solving OT, they omit detailed evaluation of this aspect in high dimensions. The experiments in 2D are good but do not convincingly support the claim, more advanced and high-dimensional evaluation should be considered [3] and some recent baselines should be included like [4].

5) Some of the theoretical statements are not very mathematically rigorous. For example, the authors prove some results regarding the optimal dual potentials (like eq. 10/11), but do not explain to which functional spaces they belong. If I correctly get it from the proof, they should be continuous functions. Does the supremum among the continuous functions is achieved, i.e., are f* also continuous functions?

References

[1] Agueh, M., & Carlier, G. (2011). Barycenters in the Wasserstein space. SIAM Journal on Mathematical Analysis, 43(2), 904-924.

[2] Kolesov, A., Mokrov, P., Udovichenko, I., Gazdieva, M., Pammer, G., Burnaev, E., & Korotin, A. Estimating Barycenters of Distributions with Neural Optimal Transport. In Forty-first International Conference on Machine Learning.

[3] Korotin, A., Li, L., Genevay, A., Solomon, J. M., Filippov, A., & Burnaev, E. (2021). Do neural optimal transport solvers work? a continuous wasserstein-2 benchmark. Advances in neural information processing systems, 34, 14593-14605.

[4] Amos, B. On amortizing convex conjugates for optimal transport. In The Eleventh International Conference on Learning Representations.

**Questions:**

I think the ideas in this paper are very interesting and should be presented to the community. My current score is based on the current condition of the paper but I may adjust it if the authors carefully reply to the weaknesses which I raised and revise the paper accordingly. Also, I have some additional questions:

1) What is the point of introducing alpha? The OT map/displacement maps should be the same for all alpha, right?

2) Could you please provide some analysis of the time sampling schemes (line 294)? In diffusion models, this is an important aspect, so I believe it may be important here as well and at least some analysis should be provided. For example, you can consider a scheme where t is mostly samples closer to 0/1 and the other scheme where t is concentrated around 0.5 and show the results.

3) It looks like the training curves (figure 5) present the losses which are generally not very representative in adversarial learning. Could you please provide FID(epoch) plots to see how stably your method converges compared to the baselines? This would be much more convincing.

4) Most comparisons are quantitative through FID which does not measure optimality but only measures matching the target. Could you please provide a side-by-side qualitative comparison with the baseline in I2I tasks? It would be nice to see how your trained generator preserves the content compared to the baselines.

5) Could you please run your method in some I2I experiment several times. Does it converge to roughly the same solutions (qualitatively), i.e., recovers (nearly) the same map (which should be optimal)?

---

> ### Author Response · Authors · 2024-11-22
> **Response to Reviewer xVqq (1/3)**
>
> We sincerely thank the reviewer for carefully reading our manuscript and providing valuable feedback. Moreover, we appreciate the reviewer for considering "the contribution of this paper looks as significant for the neural OT field". We hope our responses to be helpful in addressing the reviewer's concerns. We highlighted the corresponding revisions in the manuscript in Blue.
>
> $ $
>
> ---
>
> **W1.**
> I believe that there might be a theoretical gap in the proposed DI-OTM approach which lies in the restricted parameterization of the t-dependent transport maps. Specifically, each transport map (for a particular t) should be parameterized the way that it should solve the corresponding inner conjugation (c-transform) minimization for a particular corresponding dual potential (for time t). However, when the authors tighten all the transport maps together via a single function, this may not hold and may spoil the theoretical validity of the proposed semi-dual form. This aspect should be discussed in more detail.
>
> **A.**
> We appreciate the reviewer for the insightful comment. For clarification, as the reviewer commented, the optimality condition (Eq. 17) between the minimizer of the inner problem (c-transform) is satisfied under the optimal potential $V_{t}^{\star}$. Formally, given the optimal potential $V(t,x)^{\star}$, let $\overrightarrow{T}^{\star}$ be the optimal forward transport maps. Then, $\overrightarrow{T}\_t (x)$ becomes the minimizer of the inner-optimization problem:
> $$\overrightarrow{T}\_t (x) = (1-t) x + t \overrightarrow{T}\_\theta (x) \in {\rm{arginf}}\_{y\in \mathcal{Y}} \left[ c(x,y) - tV\_{t}^{\star}(y) \right]$$
> However, during training, the potential network $V_{\phi}$ is not optimal. Therefore, there might be a gap between theoretical guarantees and practical convergence. We included this additional discussion in the revised version of our manuscript in Line 256-260.
>
> Nevertheless, the optimal transport map $\overrightarrow{T}^{\star}$ satisfies the above displacement interpolation relationship between minimizers. Practically, to enhance training efficiency, we adopted the displacement placement interpolation for $\overrightarrow{T}_t, \overleftarrow{T}_t$.
> $$
> \overrightarrow{T}\_t (x) = (1-t) x + t \overrightarrow{T}\_\theta (x), \quad \overleftarrow{T}\_t (y) = (1-t) y + t \overleftarrow{T}\_\theta (y)
> \quad \textrm{ for } t \in (0,1).
> $$
> Note that, as discussed in Lines 250-251, **this parametrization can be understood as leveraging the optimality condition**. Therefore, this parametrization can serve as a regularization, introducing inductive biases that may help promote consistency along the time-varying transport maps.
>
> $ $
>
> ---
>
> **W2.**
> I think that some of the results presented here are not completely novel and the authors miss a large set of related work. The key problem which is exploited in the current work is the displacement interpolation optimization (equation 8). In essence, this is the Wasserstein-2 barycenter problem and, to my understanding, it has already been well studied both in theory and in practice. For example, the W2 dual barycenter problem (equation 9 in theorem 3.1 in the current paper) has been derived in the founding work [1], see their derivations around proposition 2.2. The semi-dual version (which is the second part of theorem 3.1 in the current paper) seems to directly follow from the general semi-dual for barycenters which has been recently introduced in [2] (theorem 4.1). I think these relations to the barycenter literature (theoretical and computational) should be clearly clarified and the related literature should be included.
>
> **A.**
> We agree with the reviewer that Thm 3.1 in our manuscript can be derived from the dual form of the barycenter problem. Specifically, $\mathcal{L}_{DI}$ in Eq. 8 can be interpreted as the Wasserstein barycenter problem between two probability distributions. **We clarified this point in the revised version of our manuscript** in Lines 142-146 as follows:
>
> > Note that $\mathcal{L}\_{DI}$ corresponds to the Wasserstein-2 barycenter problem between the two probability distributions $\mu, \nu$ [1, 2]. In other words, Eq. 8 represents the equivalence between the displacement interpolants and the Wasserstein barycenter. This equivalence will be utilized in Sec 3 to derive our approach to neural optimal transport, i.e., learning the optimal transport map $T^{\star}$ with a neural network. We establish how the optimal potential and transport maps for each $\rho_{t}^{\star}$ are related and use this relationship to improve neural optimal transport.
>
>
> Moreover, we would like to emphasize **two key additional contributions of our work**: (1) we established how these optimal potentials are related using the Hamilton-Jacobi-Bellman (HJB) equation (Thm 3.3) and (2) we utilized both this relationship and the displacement interpolation to achieve a more stable and accurate estimation of neural optimal transport.

---

> ### Author Response · Authors · 2024-11-22
> **Response to Reviewer xVqq (2/3)**
>
> ---
>
> **W3.**
> The DIOTM approach proposed here seems to work only for the quadratic cost optimal transport (and may be for some lp-based OT as well) due to reliance on the displacement interpolation properties. It looks like it can not be generalized to more general OT formulations, e.g., formulations with non-lp transport costs. This point is more a limitation than a weakness as the authors specifically target the quadratic cost OT. Nevertheless, it should be mentioned in the paper and the background considers the general cost OT.
>
> **A.**
> We agree with the reviewer that the assumption of the quadratic cost function is a limitation of our work. Hence, we explicitly stated this assumption in Lines 70-71 and 123-124. However, we would like to emphasize that **this quadratic cost is dominant in the OT literature across diverse machine learning applications**, such as generative modeling [1], image-to-image translation [2, 3], and predicting single-cell perturbation responses [4]. Following the reviewer's advice, we further clarified this limitation in the Conclusion section as follows:
>
> > Another limitation of this work is that our approach is limited to the quadratic cost. This is because our displacement interpolation parametrization in Eq. 17 is only valid under the quadratic cost assumption.
>
> $ $
>
> [1] Rout, Litu, Alexander Korotin, and Evgeny Burnaev. "Generative modeling with optimal transport maps." ICLR 2022.
> [2] Korotin, Alexander, Daniil Selikhanovych, and Evgeny Burnaev. "Neural optimal transport." ICLR 2023.
> [3] Fan, Jiaojiao, et al. "Scalable computation of monge maps with general costs." ICLRW 2022.
> [4] Bunne, Charlotte, et al. "Learning single-cell perturbation responses using neural optimal transport." Nature methods.
>
> $ $
>
> ---
>
> **W4.**
> While the authors claim that they significantly improve the accuracy of solving OT, they omit detailed evaluation of this aspect in high dimensions. The experiments in 2D are good but do not convincingly support the claim, more advanced and high-dimensional evaluation should be considered [3] and some recent baselines should be included like [4].
>
> **A.**
> We would like to gently remind that **our DIOTM is also evaluated on the image-to-image translation task on 128x128 resolution images** (Table 2). Given that existing approaches in the OT literature usually evaluate their methods on comparable or lower resolution images [1,2,3], the 128x128 resolution images can be considered a high-dimensional experiment.
>
> Furthermore, following the reviewer's advice, **we conducted a high-dimensional evaluation on the Gaussian mixture benchmark from [1]**. The experimental results are in Table 6 in the Appendix. Our model demonstrated reasonable performance, effectively learning the optimal transport map. Still, our model does not achieve state-of-the-art results. We believe this is because our primary goal was to achieve better scalability and stable estimation on a high-dimensional image dataset. To achieve this, our DIOTM does not impose specific structure constraints on two neural networks, parametrizing the transport map and the discriminator, such as ICNN. However, such an approach (ICNN) sacrifices scalability, particularly for image datasets.
>
> $ $
>
> [1] Fan, Jiaojiao, et al. "Scalable computation of monge maps with general costs." arXiv preprint arXiv:2106.03812 4 (2021).
> [2] Rout, Litu, Alexander Korotin, and Evgeny Burnaev. "Generative modeling with optimal transport maps." ICLR 2022.
> [3] Gushchin, Nikita, et al. "Adversarial Schr\" odinger Bridge Matching." NeurIPS 2024.
>
> $ $
>
> ---
>
> **W5.**
> Some of the theoretical statements are not very mathematically rigorous. For example, the authors prove some results regarding the optimal dual potentials (like eq. 10/11), but do not explain to which functional spaces they belong. If I correctly get it from the proof, they should be continuous functions. Does the supremum among the continuous functions is achieved, i.e., are f* also continuous functions?
>
> **A.**
> We apologize for not specifying the exact functional spaces for the potential functions. As the reviewer noted, the maximization problem over potential functions (Eq. 9 and 12) is conducted within the continuous function space, i.e., $f_{1, t}, f_{2, t} \in C(\mathcal{X}= \mathcal{Y})$. We revised our manuscript to include these functional spaces. Moreover, the continuity of the optimal potential $f^{\star}$ is achieved under proper assumptions in Caffarelli's theorem (Theorem 12.50 in [1]). Specifically, the assumptions are that $\mu, \nu$ are supported on convex connected bounded domains and are bounded above and below.
>
> $ $
>
> [1] Villani, C. (2009). Optimal transport: old and new (Vol. 338, p. 23). Berlin: springer.

---

> ### Author Response · Authors · 2024-11-22
> **Response to Reviewer xVqq (3/3)**
>
> ---
>
> **Q1.**
> What is the point of introducing alpha? The OT map/displacement maps should be the same for all alpha, right?
>
> **A.**
> $\alpha$ is a hyperparameter, which is introduced for practical purposes. Theoretically, the OT map and displacement maps should be the same for all values of $\alpha$. However, when conducting experiments on high-dimensional image datasets, selecting an appropriate $\alpha$ is necessary to make neural network training feasible.
>
> $ $
>
> ---
>
> **Q2.**
> Could you please provide some analysis of the time sampling schemes (line 294)? In diffusion models, this is an important aspect, so I believe it may be important here as well and at least some analysis should be provided. For example, you can consider a scheme where t is mostly samples closer to 0/1 and the other scheme where t is concentrated around 0.5 and show the results
>
> **A.**
> We appreciate the reviewer for the insightful comment. As a reminder, we employed uniform sampling for time $t$ in our image-to-image translation experiments. We evaluated **two alternative time sampling distributions**: the Beta distribution with $(\alpha, \beta) = (0.5, 0.5)$, which peaks at 0 and 1, and with $(\alpha, \beta) = (2, 2)$, which peaks at 0.5. The results are presented below:
>
> - Image-to-Image translation on the Wild$\rightarrow$Cat (64x64).
> |Model|Uniform|Beta (2,2)| Beta (0.5,0.5)|
> |:---|:---|:---|:---|
> |FID ($\downarrow$)| 10.72 |12.46|11.85|
>
> Interestingly, the initial uniform distribution achieved the best results. Thank you for suggesting this meaningful experiment. We incorporated this result into our Appendix (Table 5).
>
> $ $
>
> ---
>
> **Q3.**
> It looks like the training curves (figure 5) present the losses which are generally not very representative in adversarial learning. Could you please provide FID(epoch) plots to see how stably your method converges compared to the baselines? This would be much more convincing.
>
> **A.**
> **We included the FID (Training iterations) plot in Fig 14 of the Appendix**. This plot shows that our DIOTM achieves more stable FID dynamics across diverse regularization parameter $\lambda$ values. Moreover, in the original manuscript, we visualized the training loss curves to assess the stability of the adversarial training, following [1,2]. We believe that the training loss visualization can also serve as supporting evidence for the unstable training dynamics of OTM.
>
> $ $
>
> [1] Arjovsky, Martin, Soumith Chintala, and Léon Bottou. "Wasserstein generative adversarial networks." ICLR 2017.
> [2] Choi, Jaemoo, Jaewoong Choi, and Myungjoo Kang. "Analyzing and Improving Optimal-Transport-based Adversarial Networks." ICLR 2024.
>
> $ $
>
> ---
>
> **Q4.**
> Most comparisons are quantitative through FID which does not measure optimality but only measures matching the target. Could you please provide a side-by-side qualitative comparison with the baseline in I2I tasks? It would be nice to see how your trained generator preserves the content compared to the baselines.
>
> **A.**
> We included **additional qualitative comparisons of translated samples between OTM and DIOTM** in Fig 12 (Male $\rightarrow$ Female 128x128) and Fig 13 (Wiid $\rightarrow$ Cat 64x64) in the Appendix. These examples demonstrate that OTM often fails to transform target semantics correctly, e.g. resulting in Male $\rightarrow$ Male in $\lambda=50$. Furthermore, we evaluated the LPIPS score ($\downarrow)$ to quantitatively evaluate how well the generator preserves the content. The results are presented in Table 4 in the Appendix. Our model exhibits comparable LPIPS scores to the baselines.

---

### Meta-Review · Area_Chair_yecf · 2024-12-18

**Metareview:**

The reviewers agree that the paper presents a novel and interesting method for solving the optimal transport mapping problem using neural networks.  They appreciate the idea of exploiting displacement interpolation and the theoretically justified HJB-based regularization.  The experimental results on image-to-image translation are also considered convincing.

However, there are some concerns about the theoretical validity of the proposed semi-dual form and the limited scope of the method.  The reviewers also point out that some of the results are not completely novel and miss related work.

**Additional Comments On Reviewer Discussion:**

Reviewers raised concerns about theoretical limitations and suggested further high-dimensional evaluation. The authors clarified the limitations, addressed the concerns, and conducted additional evaluations.

---

### Decision · Program_Chairs · 2025-01-22

Accept (Poster)